# The role of natural gas in reaching net-zero emissions in the electric sector

**John E. T. Bistline** ⬥[1] ✉ **& David T. Young** ⬥[1]

Replacing coal with natural gas has contributed to recent emissions reductions in the electric sector, but there are questions about the near- and long-term roles for gas under deep decarbonization. In this study, we assess the potential role for natural gas and carbon removal in deeply decarbonized electricity systems in the U.S. and evaluate the robustness of these insights to key technology and policy assumptions. We find that natural-gas-fired generation can lower the cost of electric sector decarbonization, a result that is robust to a range of sensitivities, when carbon removal is allowed under policy. Accelerating decarbonization to reach net-zero in 2035 entails greater contributions from natural gas than in 2050. Nonetheless, wind and solar have higher generation shares than natural gas for most regions and scenarios (52-66% variable renewables for net-zero scenarios versus 0-19% for gas), suggesting that natural gas generation can be substituted more easily than its capacity.

The power sector is expected to play a central role in economy-wide decarbonization, both through direct emissions reductions and through end-use electrification[1,2]. Natural gas has historically contributed to emissions reductions in many regional power systems in the U.S.[3,4]. But there are questions about the near- and long-term roles for natural gas as deep greenhouse gas emissions reductions are pursued, especially net-zero targets where emissions produced from resources such as natural gas are balanced by an equivalent amount of carbon removal. Utilities are pledging net-zero targets that can include plans to build gas-fired capacity, which raises questions about levels of natural gas that are consistent with electric sector decarbonization goals.

Previous studies examine the role of natural gas in reducing emissions in the power sector[5-9]. However, these studies do not look at reaching zero emissions goals or accelerated decarbonization in line with the U.S. target of "100 percent carbon pollution-free electricity by 2035"[10]. Expected cost declines for renewables and storage and natural gas prices have evolved since earlier studies were conducted[11,12].

Our objective is to assess the potential role for natural gas and carbon removal in deeply decarbonized electricity systems in the U.S. and evaluate the robustness of this role to key technology and policy assumptions. Our analysis extends the existing literature in several ways. First, we use a detailed energy systems model to evaluate how the role of natural gas could change electric sector planning decisions and

costs in the U.S., especially under a zero-emissions goal. Second, the analysis includes a wide range of sensitivities, including assessing impacts of accelerating zero-emissions goals to 2035, per the updated U.S. Nationally Determined Contribution[10]. Third, we evaluate the role of natural gas in a range of regional power system contexts with different existing capacity mixes, natural gas prices, renewable resources, and demand characteristics. Finally, the analysis models electric sector investment and operational decisions with full hourly temporal resolution, endogenous end-use decisions and load shapes, as well as a greater suite of technological options to better represent the economic characteristics of variable renewables, energy storage (both short-duration options like batteries and longer-duration ones like electrolytic hydrogen), and dispatchable low-carbon technologies. Hourly resolution is important not only for accurately characterizing the investment and operations of electric sector resources but also for capturing sector coupling dynamics such as load flexibility and fuels production.

We find that natural gas capacity and generation can play key roles in electric sector decarbonization—both during the transition to zero and at the destination—but the extent depends on key uncertainties related to policy design, availability of carbon removal, ability to mitigate upstream methane emissions, and transition risks related to technological change. Natural gas has a role in the least-cost path in the sensitivities examined here and as part of a zero-emissions system in all cases except when policy design precludes its inclusion, findings that

[1]Electric Power Research Institute, 3420 Hillview Avenue, Palo Alto, CA 94304, USA. ✉e-mail: jbistline@epri.com

are robust to a wide range of alternate assumptions. New and existing gas-fired units can provide firm, flexible capacity that can ensure electricity demand is met in every hour as coal retires and as electrification increases demand—helping to reduce emissions, ensure system dependability, and keep transition costs low. Regions with lower quality renewable resources have higher natural gas shares, or higher costs associated with decarbonization if natural gas is unavailable. Wind and solar exhibit greater increases in generation shares for many regions and scenarios, especially with stringent $CO_2$ policies (52–66% variable renewables for net-zero scenarios versus 0–19% for natural gas). The analysis quantifies transition risks for natural gas plant developers and operators, policymakers, and other stakeholders interested in feasible and affordable electric sector decarbonization pathways to mitigate climate change.

## Results

### Modeling deep decarbonization in the electric sector

To evaluate the potential role of natural gas in deep decarbonization of the electric sector, it is important to accurately characterize the economics of complementary and competing technologies, including variable renewables, energy storage, and other firm low-emitting technologies[13,14]. Firm resources refer to "technologies that can be counted on to meet demand when needed in all seasons and over long durations"[15] such as hydropower, biomass, geothermal carbon-capture-equipped capacity, and zero-carbon gas-fueled plants (e.g., hydrogen). To appropriately represent these resources, we use a detailed energy systems model, US-REGEN, with hourly resolution, which captures the joint decisions over dispatch, energy storage charge/discharge, curtailment, hydrogen production/storage, and inter-regional transmission and $CO_2$ pipeline flows, and in turn the implications of these hourly decisions for the economics of new investments. Carbon removal is represented through bioenergy with carbon capture and storage (BECCS) and direct air capture (DAC).

Sector coupling, including fuels production and the opportunity to defer electric vehicle charging to reduce or defer peak demand, is also represented. US-REGEN is documented in detail in EPRI (2020)[16], so only summaries of key features are provided in the "Methods" and Supplementary Information (Supplementary Notes 1 and 2).

The analysis considers three policy targets:
- Reference, including significant on-the-books federal and state electric sector policies and incentives as of June 2021 but no additional policies, as described further in the Methods section;
- Carbon-Free, includes all policies in the Reference and requires that no $CO_2$-emitting technologies be operating beginning in the target year, which is implemented as a constraint on national electric sector greenhouse gas emissions (including direct $CO_2$ emissions and upstream $CH_4$), corresponding to the emissions trajectories in Supplementary Fig. 3 in Supplementary Note 2; and
- Net-Zero, includes all policies in the Reference and requires that any remaining $CO_2$ emissions from the electric sector in or after the target year to be balanced by sequestration such that no additional $CO_2$ is added to the atmosphere.

For the deep decarbonization scenarios, we consider target years of:
- 2035 (aligning with the updated U.S. Nationally Determined Contribution goal "to reach 100 percent carbon pollution-free electricity by 2035"[10]); and
- 2050 (including an interim goal of 80% below 2005 levels by 2035).

In addition to these targets, we consider sensitivities to other technology, market, and policy assumptions that might significantly alter the economics of natural gas versus other technologies. These sensitivities are summarized in Table 1 and discussed in detail in Supplementary Note 2.

**Table 1 | Summary of scenarios**

| Scenario (Abbr.) | Description |
|---|---|
| **Policy targets** | |
| Reference | On-the-books federal and state electric sector policies and incentives |
| Net-zero | Net carbon emissions equal zero nationally |
| Carbon-free | Electricity generation does not use fossil fuels or does not emit carbon |
| **Policy timeframe** | |
| 2035 | Zero emissions target by 2035 |
| 2050 | Zero emissions target by 2050 |
| **Natural gas price projections** | |
| Low | U.S. EIA *Annual Energy Outlook* High Oil and Gas Supply |
| Reference | U.S. EIA *Annual Energy Outlook* Reference |
| High | U.S. EIA *Annual Energy Outlook* Low Oil and Gas Supply |
| **Technology and policy sensitivities (assuming net-zero by 2035 target)** | |
| Reference (NZ ref) | N/A |
| Lower renewables and battery costs (LoRE) | Capital costs for wind, solar, and batteries exhibit faster declines (Supplementary Fig. 5) |
| Zero emission fossil CCS (HiCapture) | Availability of a CCS-equipped gas technology where the flue gas has $CO_2$ concentration similar to the atmosphere and costs similar to 90% capture |
| No new NGCC capacity (NoGas) | No new NGCC capacity investment allowed in any region after 2020 |
| No new NGCC or CCS capacity (NoGasCCS) | No new NGCC or CCS-equipped capacity allowed in any region after 2020 |
| Upstream methane with 3% leakage (3% Leak) | Adjust upstream gas system $CH_4$ releases with 3% leakage rate (instead of the reference assumption of 1.5%) |
| CCS tax credits (45Q) | Section 45Q tax credits of \$32/t-$CO_2$ for sequestered $CO_2$ in 2020 increasing to \$50/t-$CO_2$ by 2026 |
| Low-cost long-duration energy storage (LDES) | Stylized long-duration storage availability with energy capacity costs of \$10/kWh, consistent with the U.S. DOE's Long Duration Storage Shot |
| Pessimistic natural gas assumptions (Pess) | Combining pessimistic assumptions about gas (high $CH_4$ leakage, high prices, high BECCS cost, no DAC, and high $CO_2$ storage costs) and optimistic renewables, storage, and electrolyzer costs |

Detailed descriptions in the "Methods" section and Supplementary Note 2. Combinations of the different classes of sensitivities are conducted.

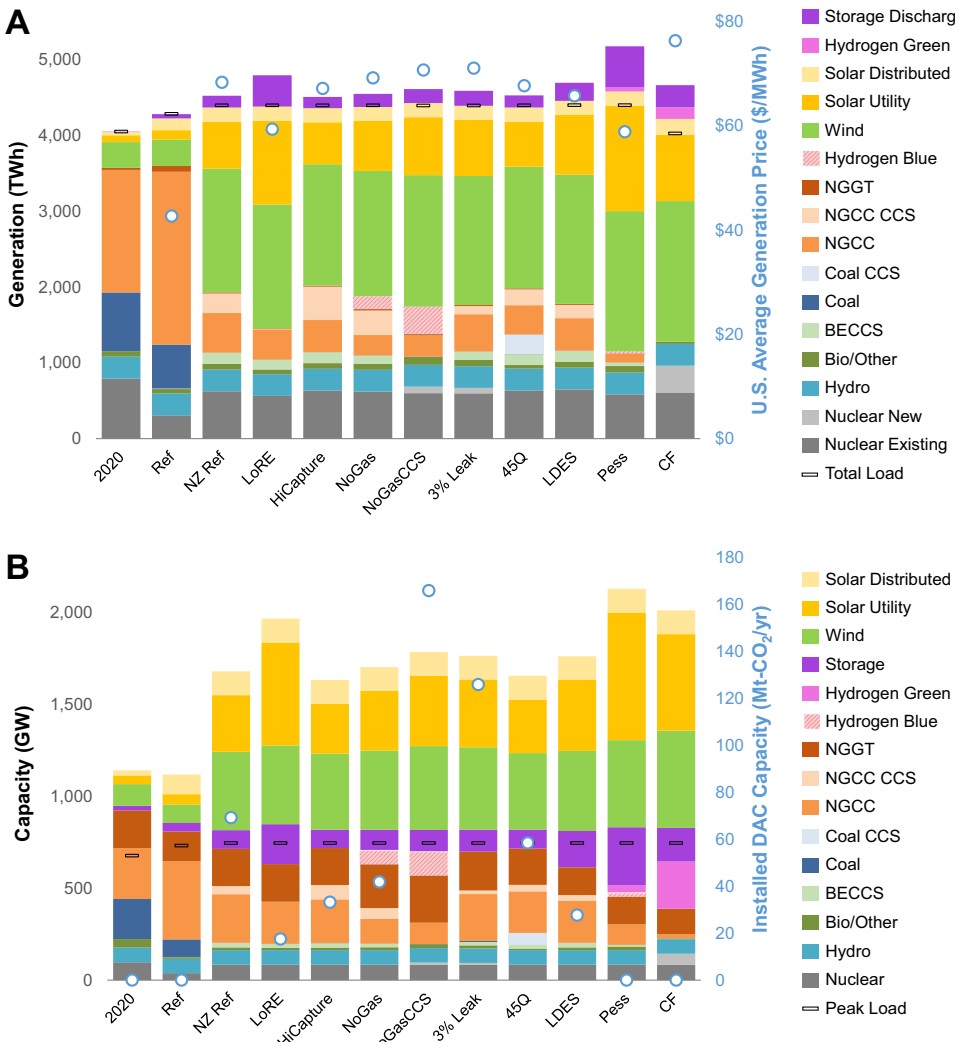

**Fig. 1 | National generation and capacity by technology and scenario in 2035.**
**A** Generation in 2035 assuming net-zero-emissions targets in 2035. U.S. generation-weighted average price ($/MWh), which reflects all generation, new bulk transmission, carbon removal, and $CO_2$ transport and storage costs. Storage Discharge refers to gross discharge from non-hydrogen technologies. **B** Capacity of generation, energy storage, and direct air capture (DAC) in 2035. Scenario definitions and abbreviations are provided in Table 1. NGGT natural gas turbines, NGCC natural gas combined cycle, CCS carbon capture and sequestration, BECCS bioenergy with CCS.

Natural gas price assumptions come from the U.S. Energy Information Administration's *Annual Energy Outlook* reference case with high and low price sensitivities[17]. Biomass fuel costs are represented as regional supply curves, which are based on the Forest and Agriculture Sector Optimization Model with Greenhouse Gases (FASOM-GHG) (Supplementary Fig. 10).

Annual electricity demand and hourly load shapes are generated endogenously by the REGEN end-use model (as described in Methods and Supplementary Note 1). To reflect the deep decarbonization context for the Net-Zero and Carbon-Free scenarios, the end-use model assumes $CO_2$ pricing of $50/t-$CO_2$ for non-electric sectors beginning in 2025 that increases at 7% per year, which is intended as a proxy for a suite of $CO_2$ policies for end-use sectors. We conduct sensitivities across different levels of end-use electrification to examine effects on natural gas deployment.

### Drivers of natural gas use in electric sector decarbonization strategies

Natural gas generation and capacity (Fig. 1, top and bottom panels, respectively) are robust elements of least-cost decarbonization portfolios, not only during the transition to net-zero emissions but also at the destination. Lower-emitting firm resources are valuable in systems

with high renewable penetration to balance variability across weeks and seasons (Supplementary Fig. 17) and to replace retiring coal capacity, and we find that natural gas paired with capture or $CO_2$ removal is the cheapest form of low-emitting capacity for many U.S. regions and scenarios except when new natural gas capacity is not allowed. Supplementary Fig. 13 shows how coal generation shares rapidly decline across all regions in emissions-constrained scenarios, while natural gas shares exhibit much slower declines. The generation from new natural gas also has value for decarbonization as coal retires and electrification increases load (Supplementary Fig. 27), which are key elements of economy-wide decarbonization[18].

The extent of natural gas generation for a decarbonized electric sector depends on policy design, ability to mitigate upstream methane, and transition risks from technological change (Fig. 1). Demand for natural gas capacity varies from 160 to 590 GW across the Net-Zero scenarios (Fig. 1, bottom panel). Natural gas deployment is lower if there are constraints on new builds (NoGas, NoGasCCS, CF), if renewables costs are lower than expected (LoRE), if there is a breakthrough in long-duration energy storage (LDES), if other clean firm capacity is subsidized (45Q), or if a combination of these drivers occurs (Pess). Natural gas deployment is higher if methane leakage is low (NZ Ref) and if a higher carbon capture and sequestration (CCS) capture

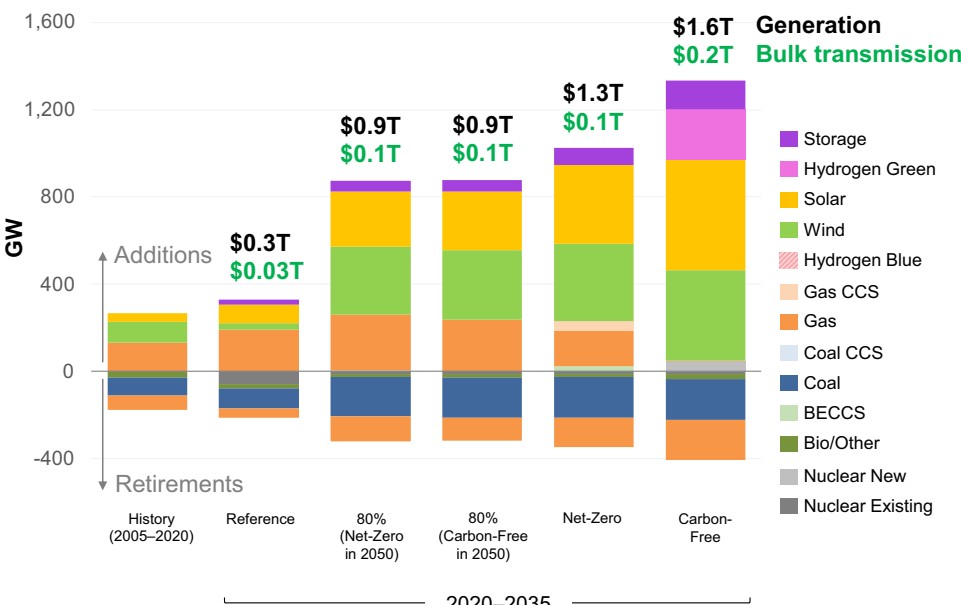

**Fig. 2 | Capacity investments and retirements by technology and scenario through 2035.** Historical additions and retirements come from Form EIA-860 data. Monetary investments in generation and bulk transmission are shown above each bar and include cumulative capital expenditures. CCS carbon capture and sequestration, BECCS bioenergy with CCS.

rate technology is available (HiCapture). As the choice set of firm technologies is constrained or made more costly, the generation and capacity mixes move closer to the Carbon-Free (CF) scenario, where carbon removal and natural gas are prohibited.

In these scenarios, carbon dioxide removal (CDR) comes from BECCS, which provides flexibility to balance renewables and allows for limited positive emissions from natural gas, hydrogen production, and other system resources[19], and DAC. BECCS is the main CDR technology used to reach net-zero goals due to its lower cost of net $CO_2$ removal and provision of firm negative-$CO_2$ electricity; however, DAC deployment increases under alternate technological cost and availability assumptions (Supplementary Fig. 22). A net-zero policy might require emitting resources to purchase CDR credits, which would raise the dispatch costs of natural-gas-fired generators. BECCS is modest in generation and capacity terms, but its negative emissions are roughly three times as large as the positive emissions from natural gas combined cycle (NGCC) without CCS (the reference Net-Zero scenario has about 150 TWh of BECCS and 520 TWh NGCC). The scenario that prohibits new gas additions or CCS-equipped capacity (NoGasCCS) leads to greater production and generation from blue hydrogen, which uses direct air capture to offset residual emissions from natural gas and hydrogen production.

Wind and solar exhibit greater increases in generation shares than natural gas for many regions and scenarios (52% to 66% wind and solar for Net-Zero scenarios versus 0% to 19% for natural gas). Natural gas has a larger share of capacity than generation for Net-Zero scenarios, ranging from 8% to 32% (Fig. 1, bottom panel). Note that NGCC units with carbon capture and sequestration (CCS) play a different role than uncaptured gas. The former is a higher capital cost and higher capacity factor option but is only deployed up to a point versus wind and solar, after which uncaptured gas of some form is preferred, either NGCC or peaking units.

Accelerating decarbonization entails greater contributions from gas on a relative and absolute basis (Fig. 6), as capital costs of other low-emitting technologies are assumed to fall through 2050. Targeting zero emissions by 2035 (instead of 2050) in the electric sector entails lower deployment of solar and battery storage and higher CCS-equipped natural gas, wind, and new nuclear (Supplementary Fig. 18).

Note that solar generation shares are lower than some earlier U.S. decarbonization studies (e.g. refs. 20, 21), due to our modeling having higher temporal resolution, accelerated decarbonization, and endogenous end-use decisions with hourly load shapes and electrification, which can be associated with lower solar deployment vis-à-vis wind and other low-emitting generation options[14,22,23]. Figures aggregate onshore wind and offshore wind into a single category, since offshore wind capacity is driven primarily by state mandates (approximately 32 GW by 2035) and does not vary considerably across scenarios. Supplementary Figs. 27 and 28 show impacts of electrification on electricity demand and hourly load shapes across these scenarios, which can lead to shifting peak loads toward winter during periods with low solar output in some regions.

## System value of natural gas

One method of quantifying the system value of gas is to compare investment and cost outcomes for Net-Zero decarbonization (i.e., without technology restrictions) and a Carbon-Free scenario (i.e., where generation from natural gas and CCS are prohibited). Figure 2 shows investments and expenditures across these decarbonization scenarios. Reaching zero emissions by 2035 entails an exceptional and unprecedented scale of changes. We find that the Carbon-Free scenario requires over 1300 GW of cumulative investments in solar, wind, new nuclear, hydrogen, and battery storage by 2035 with generation expenditures of $1.6 trillion. The Net-Zero scenario enables the use of existing and new gas, which lowers cumulative investments by 2035 to about 1000 GW and also lowers generation and transmission expenditures to $1.3 trillion (see also Supplementary Fig. 12 for generation changes).

Another metric to assess the system value of natural gas is electricity prices. $CO_2$ targets, timetables, and technological assumptions all impact electricity prices (Fig. 3). For decarbonization by 2035, electricity price increases vary by region and span 45% to 104% under the Carbon-Free scenario (relative to the Reference scenario), falling to 38% to 80% in the Net-Zero scenario. In the Carbon-Free scenario, electricity price increases disincentivize electrification and hence increase non-electric gas demand (Supplementary Fig. 31). Note that this emissions rebound effect could be mitigated if a quantity-based economy-wide emissions policy (e.g., cap-and-trade) were used as the

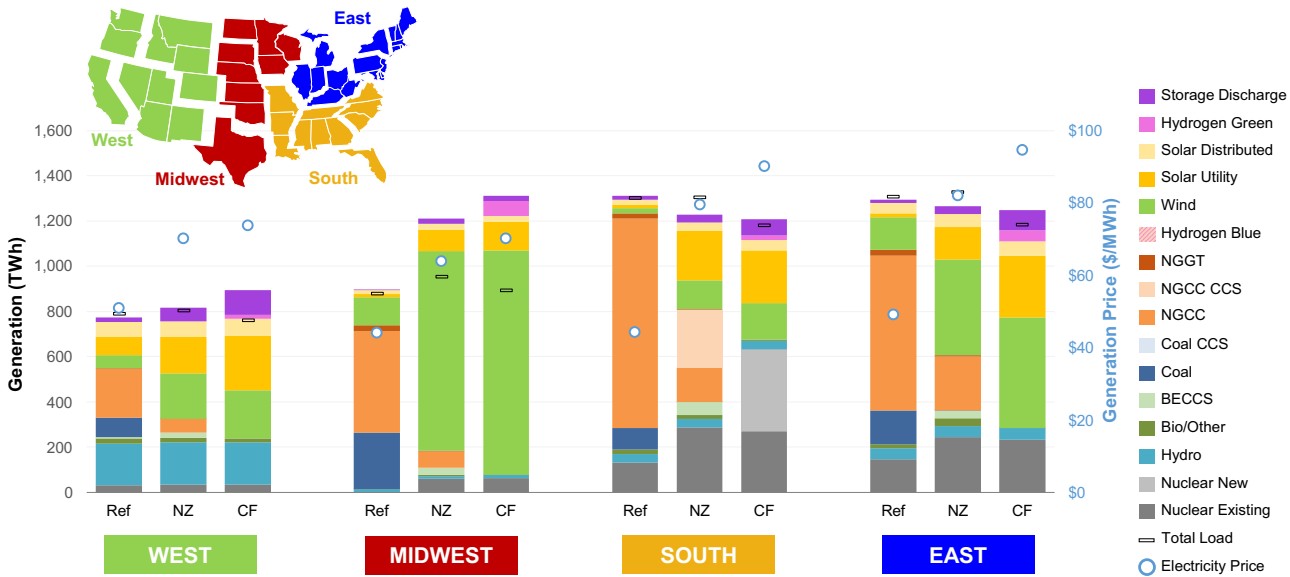

**Fig. 3 | Regional implications of policy targets on generation and electricity prices.** Results are shown for the three policy scenarios (Ref Reference; NZ Net-Zero; CF Carbon-Free). The generation-weighted average price ($/MWh) reflects all generation, new bulk transmission, carbon removal, and $CO_2$ transport and storage costs. Estimates reflect electric sector system costs as a proxy for customer impacts and do not include climate damages or other social costs or benefits. NGGT natural gas turbines, NGCC natural gas combined cycle, CCS carbon capture and sequestration, BECCS bioenergy with CCS.

primary policy instrument instead of a price-based $CO_2$ policy. A 2050 zero-emissions target allows for a more gradual introduction of new technology at lower assumed capital costs (Supplementary Fig. 4) and thus lowers electricity prices. However, note that electricity prices only track system costs and do not explicitly include monetized estimates of climate damages avoided from lower $CO_2$ emissions or co-benefits such as human health benefits from air quality improvements[24], which would be higher under nearer-term decarbonization pathways. Technological cost and availability assumptions also shape electricity prices (Fig. 1, top panel), but these differences are generally smaller than policy-related ones.

Policy stringency and the value of natural gas and carbon removal also can be evaluated by comparing shadow prices on the $CO_2$ emissions cap constraint. Marginal abatement costs increase sharply as zero emissions without CDR are approached, which are approximately $48,000/t-$CO_2$ by 2035 in the Carbon-Free scenario. With CDR in the Net-Zero scenario, marginal abatement costs corresponding to the scenarios in Fig. 1 range from $107/t-$CO_2$ (LoRE) to $126/t-$CO_2$ (Pess), which is the marginal cost of capture from BECCS.

The value of natural gas shifts from providing energy to providing capacity over time and with deeper decarbonization. As emissions decline and renewable penetration increases, natural gas capacity factors decline by roughly 50% from current levels, as NGCC units are deployed as peakers rather than as baseload suppliers (Supplementary Fig. 16). Capacity factors are generally higher for CCS-equipped natural gas rather than unabated NGCC capacity— 55–75% versus 0–70%, respectively, across different regions and scenarios (Supplementary Fig. 16). Moving from a reference to a net-zero policy environment lowers returns to existing NGCC capacity and shifts its value from bulk electricity sales to firm capacity (Supplementary Fig. 19). The magnitude of this asset impairment varies considerably by region. Supplementary Fig. 20 illustrates how net profitability of natural gas is driven by its firm back-up role for renewables and how a large fraction of its revenues occurs in the first 15 years of operation, suggesting that near-term gas builds can be profitable even when long-run policies limit their eventual contribution. Although levelized costs of solar are lower than operating costs of gas units, normalized revenues are higher for natural gas capacity, which is why new natural gas additions can be economic as

utilization declines with deeper decarbonization (Supplementary Fig. 21).

## Regional differences in decarbonization strategies

There are important differences in the competitiveness of natural gas across U.S. regions (Fig. 3). We find that regions with lower quality renewable resources have higher natural gas shares (Fig. 4) and incur higher costs if decarbonizing without natural gas (Fig. 3). The East and South regions (both with comparatively lower wind and solar resource quality) are most impacted by target definitions, but technology eligibility affects the generation mix and trade for all regions. These regions have the highest natural gas use under Net-Zero policies and exhibit the highest increases in electricity prices under the Carbon-Free scenario. The South exhibits the highest NGCC with CCS in the Net-Zero scenario and nuclear in the Carbon-Free scenario.

Natural gas generation displaces coal generation, which declines rapidly across all regions under deep decarbonization policies (Supplementary Fig. 13). However, the rate of gaseous fuel decline exhibits regional differences due to variation in renewable resource quality, fuel prices, existing capacity mixes, and state-level policies.

Figure 4 shows how regional use of gas is highest in scenarios with lower gas prices, with a 2035 target, and those with flexible policy targets. We observe regional variation in the responsiveness to higher and lower natural gas prices. Policy assumptions have a greater influence on the competitiveness of natural gas than do fuel prices (Supplementary Fig. 26). Targeting net-zero emissions in 2050 leads to lower natural gas generation shares than net-zero in 2035 due to the lower costs of renewables and energy storage over time. Even with a Carbon-Free target in 2050, gas generation is a part of the least-cost mix in 2035 for many regions.

Companies have announced plans to cofire or blend hydrogen at existing and new natural-gas-fired plants or to fully convert these plants in the future, and gas turbine manufacturers are designing equipment to handle large shares of hydrogen[25]. However, hydrogen generation shares are modest across many scenarios in this analysis due to their higher marginal abatement costs. For instance, for $1/kg hydrogen (roughly the current costs of production with steam methane reforming or 2050 costs with electrolysis according to BloombergNEF[26]) and $4/MMBtu natural gas, abatement costs of

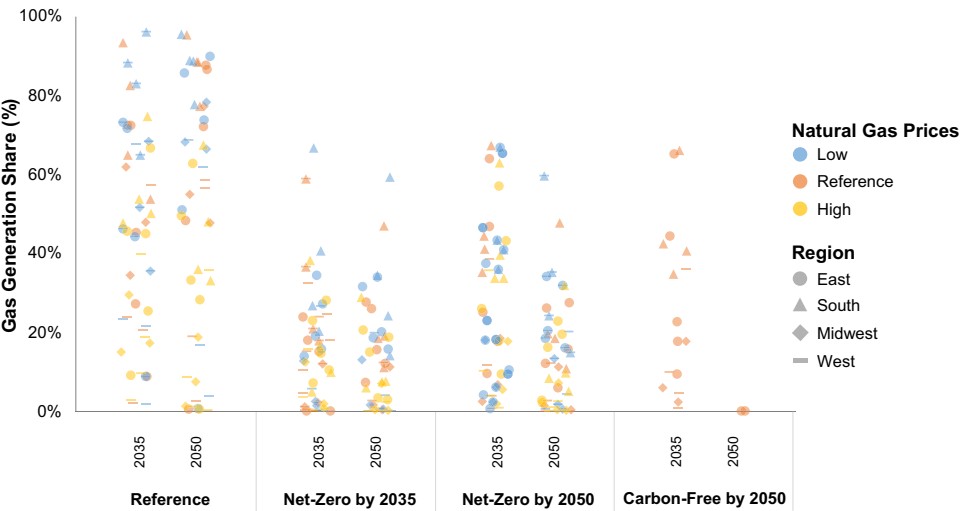

**Fig. 4 | Regional generation shares by policy scenario in 2035 and 2050.** Shares include natural-gas-fired generators with and without carbon capture as well as hydrogen through steam methane reforming. Natural gas price sensitivities are shown by color, and regions are shown by shape (regional definitions are shown in Fig. 3). Note that points are jittered horizontally within a column for greater visibility of individual values.

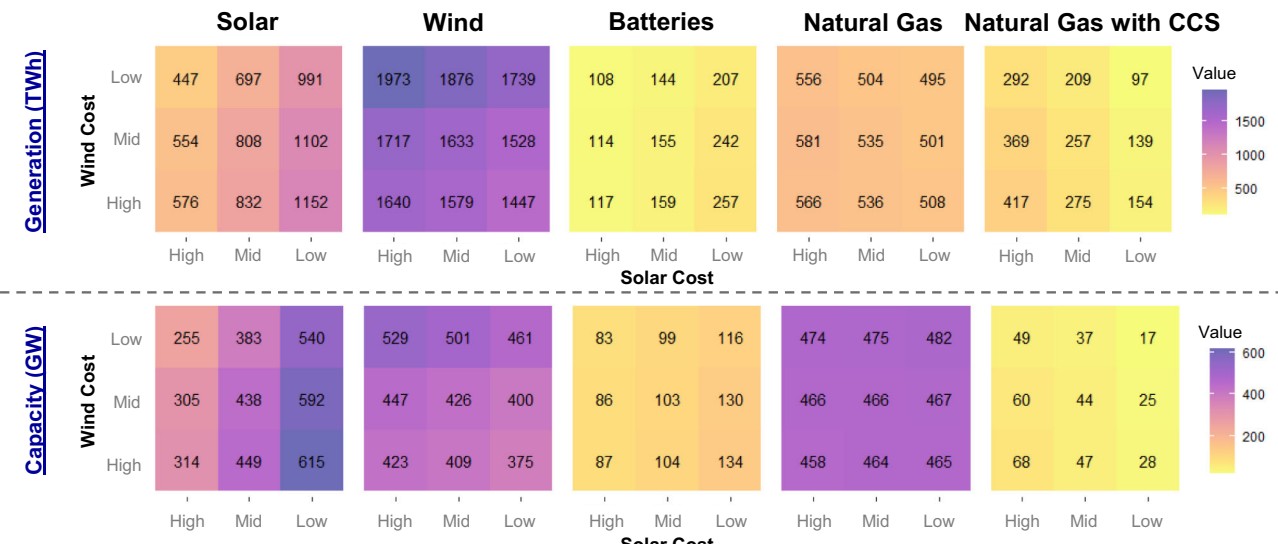

**Fig. 5 | Generation and capacity by technology (columns) across wind/solar cost scenarios (panels) in 2035.** Results are shown for the Net-Zero in 2035 scenario. Low, Mid, and High renewable cost assumptions are shown in Supplementary Fig. 5. CCS carbon capture and sequestration.

hydrogen cofiring, blending, or conversion are about $90/t-CO_2 before accounting for upstream emissions associated with hydrogen production[27]. At these marginal abatement costs, power sector $CO_2$ emissions can be lowered 90-95% from 2005 levels[19], indicating that natural gas to hydrogen fuel switching would not be economic unless lower relative costs were achieved.

## Sensitivity to wind and solar costs

Given the uncertainty about future cost declines for renewables, we run sensitivities where Low, Mid, and High costs for wind and solar (Supplementary Fig. 5) are varied independently to examine impacts on the deployment of natural gas and other technologies. Figure 5 shows generation and capacity for different technologies across these renewable cost scenarios. Unsurprisingly, the extent of solar and wind deployment depends on their own future cost declines. Battery deployment primarily varies based on the solar cost assumptions given the complementarity between these two resources. However, lower renewable costs and higher deployment do not necessarily guarantee

large markets for energy storage, as high penetrations of renewables are possible even where battery deployment is modest in capacity terms.

Unabated natural gas capacity is relatively high across these scenarios, but capacity factors and generation from these resources are low as is variation across wind and solar costs. The model finds unabated natural gas capacity (paired with carbon removal to offset emissions) to be the cheapest form of firm capacity (Fig. 1B), especially for providing dispatchable capacity during high residual load periods. Natural gas with CCS is more sensitive to wind and solar costs, as CCS-equipped generation spans by a factor of four (97 TWh/year with Low costs and 417 TWh/year with High costs).

## Impact of alternate levels of end-use electrification

Earlier scenarios assume end-use electrification driven by technological change and $CO_2$ pricing. In addition to these scenarios with reference cost and performance assumptions for electric technologies (RefEl), we conduct a sensitivity to examine effects of higher end-use electrification

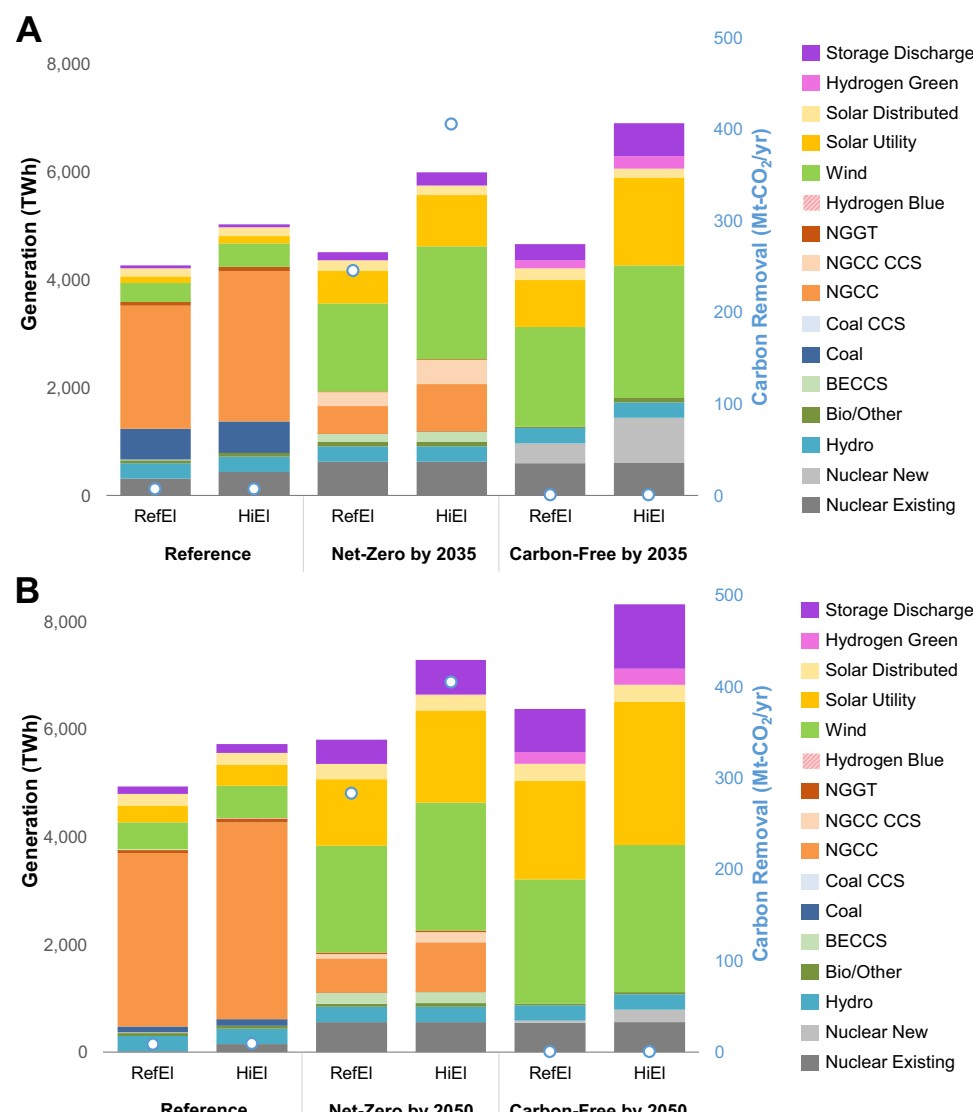

**Fig. 6 | National generation and carbon removal by technology and scenario.** Reference electrification (RefEl) and high electrification (HiEl) scenarios are shown across different electric sector policy scenarios. **A** Generation in 2035 assuming net-zero-emissions targets in 2035. **B** Generation in 2050 assuming net-zero-emissions targets in 2050. NGGT natural gas turbines, NGCC natural gas combined cycle, CCS carbon capture and sequestration, BECCS bioenergy with CCS.

on natural gas deployment (HiEl), where accelerated technological change and more stringent emissions policy increase electricity demand. Supplementary Fig. 27 in Supplementary Note 4 shows electricity demand by end-use application over time across different combinations of electrification assumptions and electric sector policies, and hourly load shape impacts are illustrated in Supplementary Fig. 28 after accounting for deferrable electric vehicle charging.

The impact of higher electrification on the generation mix varies by electric sector policy scenario (Fig. 6). Under the Net-Zero and Carbon-Free scenarios, higher electrification brings additional generation from solar, wind, natural gas, and nuclear in 2035. However, by 2050, the additional electrification and falling costs of solar and batteries bring larger responses from these technologies in both relative and absolute terms. Natural gas generation and carbon removal are lower in 2050 not only due to the increased competitiveness of solar and energy storage but also to the additional electrification and load shape flexibility.

The portfolio of CDR technologies deployed depends on technological cost and availability assumptions as well as the scale of demand (Supplementary Fig. 15). BECCS is deployed over DAC through

200–300 Mt-CO$_2$/year; however, increasing biomass costs make DAC favorable at the margin for higher demand scenarios (e.g., reaching net-zero with high electrification).

## Discussion

This analysis demonstrates how natural gas capacity and generation have roles in the least-cost decarbonization path in the sensitivities examined here and in the zero-emissions system in all cases except when policy design precludes its inclusion, findings that are robust to a wide range of alternate assumptions. Natural gas capacity can be compatible with net-zero emissions goals if a net-zero policy framing allows units with CO$_2$ removal offset. Analyses assuming natural gas plants will become stranded assets in a zero-emissions future often make implicit assumptions about policy design, such as assuming that policies will adopt a Carbon-Free framing rather than a Net-Zero one. Such binary framings also often overlook more nuanced conditions of asset impairment such as when returns to existing capital are lower but not necessarily low enough to induce retirement (Supplementary Fig. 19). Our results indicate that new and existing natural gas–enabled by carbon removal via BECCS or DAC as part of a net-zero electric

sector—can help to reduce emissions, facilitate dependable system operations, and reduce the cost of decarbonizing the electric sector. Restricting technological options—specifically natural gas with and without CCS—increases cost of electric sector decarbonization. Natural gas' role in a net-zero system hinges on carbon capture, either directly (through gas with CCS) or indirectly (through carbon removal to offset unabated gas or through CCS for blue hydrogen).

The extent of deployment and utilization of natural gas depends on policy, technology, and market uncertainties. Wind and solar exhibit greater increases in generation shares for many regions and scenarios, especially with stringent $CO_2$ policies (52–66% variable renewables for Net-Zero scenarios versus 0–19% for natural gas). These findings generally agree with earlier studies of U.S. decarbonization[20,21,28–31]—albeit this study looks at deeper decarbonization goals, lower renewable costs, and greater variety of sensitivities using a model with hourly temporal resolution and endogenous load shapes. These differences generally mean that deployed natural-gas-fired capacity is on the lower side of existing multi-model deep decarbonization scenarios[9] and economy-wide net-zero studies[21,32] due to the greater number of scenarios investigated here (including some with pessimistic assumptions about gas and optimistic ones about other technologies).

These insights focus on the potential role of natural gas in the U.S. electric sector. Several unique features about the U.S. setting may make insights less transferrable to other country contexts, including its lower-cost fossil fuel resources, plentiful biomass, high-quality wind and solar resources, and ample $CO_2$ sequestration nationally. Each of these features exhibits regional heterogeneity across the country that can give rise to variation in the competitiveness of natural gas (Fig. 4). Countries like those in the European Union not only have higher costs for natural gas and other fuels but also may see more limited CCS deployment due to infrastructure challenges associated with $CO_2$ transportation and storage.

This analysis points to several areas for future work. First, the analysis did not explicitly model operational constraints (e.g., inertia) or detailed ancillary services markets, which are services that gas units contribute to today. It is unclear how adding these features would impact investments in natural gas or other technologies, but such considerations become more important as shares of inverter-based resources grow and existing capacity retires. Second, the higher variability in hourly electricity prices and dependence on a limited number of hours to provide larger shares of revenues for natural gas and firm capacity raise questions about market design for high renewables and deep decarbonization scenarios. Finally, these scenarios examined zero emissions power sector goals in the context of deep economy-wide reductions, but overall greenhouse emissions do not reach net-zero levels by 2050. Future work should examine the role of natural gas across net-zero economy-wide futures.

## Methods
### Electric sector and energy system model
To examine the role of natural gas in deep decarbonization, this analysis uses EPRI's U.S. Economy, Greenhouse Gas, and Energy (US-REGEN) model, which features an electric sector capacity planning and dispatch model linked to an end-use model with technological, temporal, and spatial detail. US-REGEN is fully documented in EPRI (2020)[16], so only summaries of key features and assumptions are provided here.

The electric-sector model is formulated as a linear program that minimizes the net present value of total system costs subject to technical and economic constraints under given scenario assumptions about policies, technologies, and markets. This model includes endogenous capacity planning and dispatch with joint investment decisions in generation, energy storage, transmission, hydrogen production, and

$CO_2$ removal, storage, and pipelines. The US-REGEN electric-sector model was built to capture the unique economic and operational characteristics of variable renewables, energy storage, and dispatchable low-emitting technologies as well as the policies that support them[13,33].

US-REGEN represents a broad range of existing and emerging generation technologies. Three utility-scale solar photovoltaic technologies are included (fixed tilt crystalline silicon, single-axis tracking, and double-axis tracking), as well as concentrated solar with endogenous thermal storage and rooftop solar. Onshore and offshore wind are represented with hub heights ranging from 80 to 140 m. Other zero-emitting technologies include geothermal, nuclear (including generation III+ and small modular reactors), and hydrogen-fired units. Finally, US-REGEN has a robust representation of thermal units: coal-, gas-, and biomass-fired with or without carbon capture and storage. The model allows for endogenous conversions of existing coal units to gas or biomass, and for retrofits with CCS.

REGEN represents a variety of energy storage technologies such as batteries, compressed air energy storage, existing pumped hydro, and hydrogen via electrolysis. For batteries, charging and discharging capacities of the inverter are assumed to be equal, and the model endogenously selects battery storage investment and system configurations (i.e., ratio of energy capacity to power capacity) based on cost structure assumptions from ref. 34. Energy capacity and power capacity are endogenously optimized for all energy storage technologies in the model. US-REGEN includes energy storage market participation for energy arbitrage, capacity value, ancillary services (namely, operating reserves when specified), and inter-regional transmission deferral. For hydrogen storage pathways, the model independently optimizes the capacity of hydrogen production via electrolysis, hydrogen storage, and generation from hydrogen turbines. The assumed electrolysis capital costs of $200/kW are at the lower range of current estimates; the cost of electricity input is endogenously determined from the grid mix. Costs of hydrogen storage are assumed to be $50/MMBtu, which are similar to storage cost estimates for salt caverns[35].

Technological cost and performance estimates come from the literature, EPRI's Technology Assessment Guide, for which a high-level summary is publicly available via EPRI's Integrated Technology Generation Options report[36], and expert elicitations. Capital costs from 2020 though 2050 are summarized in the Supplementary Information (Supplementary Fig. 4). In addition to new investments, the model includes existing capacity endowments of pumped hydropower, conventional hydropower, nuclear, and inter-regional transmission capacity. Data to characterize the existing fleet was procured from ABB Energy Velocity.

Hourly regional renewable output and resource potentials are based on analysis and data by EPRI, AWS Truepower, and NASA's MERRA-2 dataset and provide synchronous time-series values with load. Hourly profiles used in the model solution are based on a single representative year (2015 for these experiments), and the same underlying meteorology and temperatures are used in the end-use model to develop hourly load shapes (e.g., for electric space heating in residential and commercial buildings) to avoid dampening variance through multi-year averaging.

Cross-regional exchange of electricity in a given hour is constrained by net transfer capacities of transmission between regions, which can change over time as new investments are made. Base year inter-regional transmission capacity comes from the National Renewable Energy Laboratory's ReEDS model. Transmission between regions can be endogenously added with an assumed cost of $3.85 million per mile for a notional high-voltage line to transfer 6,400 MW of capacity. Interconnection costs for utility-scale wind (solar PV) are $250/kW ($100/kW) across all regions.

Emissions factors do not include lifecycle-related emissions with generation technologies or fuels.

The emissions intensity of natural gas includes upstream gas system $CH_4$ releases with a 1.5% leakage rate (i.e., volumetric fraction of consumed natural gas). This 1.5% rate is similar to the EPA's Inventory of U.S. GHG Emissions and Sinks and bottom 95% of sites from a recent study[37]. A 100-year Global Warming Potential of 25 is used based on the UNFCCC standard used by the U.S. EPA.

The US-REGEN end-use model captures intensive and extensive margin responses through investment and operational decisions. The model focuses on representing drivers for electrification from customer and firm perspectives, with considerable heterogeneity across households, industries, and regions, and is unique in its simulating economic and behavioral factors for end-use decisions rather than specifying adoption decisions exogenously as model inputs[38]. Non-electric sectors are assumed to face a carbon price as a proxy for decarbonization incentives at the end-use level, which starts at \$50/t-$CO_2$ in 2025 and increases at 7% per year. The end-use module includes a module to value and project investments in residential and commercial rooftop solar, and another module that allows for the opportunity to defer electric vehicle charging to reduce or defer peak demand. The participation share in charging flexibility programs is assumed to be 50% for residential households and 80% for workplace charging for this analysis. The end-use model generates hourly electricity load profiles by region which are inputs to the electric sector model.

The detailed electric sector capacity expansion and end-use models are run iteratively, with the electric model passing hourly electricity prices to the end-use model, and the end-use model passing back hourly load shapes and load growth, until energy prices and demands converge between the two models (Supplementary Fig. 2). In addition to electricity load shape flexibility, there is sector coupling between electricity and fuels production, including endogenous hydrogen production with range of pathways, as described in Supplementary Note 2.

## Scenario definitions and input assumptions

The analysis considers three policy targets:

- Reference, including significant on-the-books federal and state electric sector policies and federal incentives as of June 2021, excluding the 45Q tax credit for $CO_2$ storage which is the subject of a separate sensitivity scenario. Represented policies include:

  - Renewable portfolio standards in AZ, CA, CO, CT, DC, DE, IA, IL, MA, MD, ME, MI, MN, MO, MT, NC, NJ, NM, NV, NY, OR, PA, RI, TX, VA, VT, WA, and WI (based on DSIRE, NCSL)[39], including solar carveouts in AZ, CO, DC, DE, IL, MA, MD, MN, MO, NC, NJ, NM, NV, PA, and TX.
  - Clean electricity standards in CA, MA, NM, NV, VA, and WA. All except California define clean as renewables plus nuclear, and in some cases, limited biomass. All states also have significant renewable portfolio standards.
  - Offshore wind mandates in CT, MA, MD, ME, NJ, NY, RI, and VA based on legislation in those respective states.
  - Electricity storage mandates in CA, NJ, NY, and VA.
  - California AB32, represented as a carbon tax based on projections by the California Air Resources Board[40].
  - Current Clean Air Act Section 111(b) regulations effectively prohibiting the construction of new coal-fired units without CCS.
  - Investment tax credit for solar, modeled as 30% for units built before 2020, declining to 10% for units built after 2025.
  - Production tax credit for wind, modeled as ten years of credits per MWh generated, with the credit value declining

for units completed before 2020 to units completed before 2025. No credit for units completed after 2025.
  - Regional Greenhouse Gas Initiative (RGGI), based on the 27th RGGI $CO_2$ Cap.
  - Effective prohibition on new nuclear in CA, CT, IL, MA, ME, MN, NJ, OR, and WV based on an NCSL dataset[41]. When aggregating states to the 16 regions used here, this is impactful only in California.

- Carbon-Free, includes all policies in the Reference and requires that no $CO_2$-emitting technologies be operating at and after the target year; and
- Net-Zero, includes all policies in the Reference and requires that any remaining $CO_2$ emissions from the electric sector in or after the target year to be balanced by sequestration such that no additional $CO_2$ is added to the atmosphere.

For the deep decarbonization scenarios, we consider target years of

- 2035 (aligning with the updated U.S. Nationally Determined Contribution[10] goal "to reach 100 percent carbon pollution-free electricity by 2035"); and
- 2050 (including an interim goal of 80% below 2005 levels by 2035).

In addition to these targets, we consider sensitivities to other technology, market, and policy assumptions that might alter the economics of natural gas versus other technologies. Descriptions of the scenarios are provided in Supplementary Note 2. All other model input assumptions are held constant across scenarios.

Two alternate electrification scenarios are run to explore how end-use demand affects the role of natural gas under different electric sector policy scenarios:

- Reference electrification (RefEl): The core scenarios of the analysis assume reference end-use technological and behavior assumptions, as documented at: https://us-regen-docs.epri.com
- High electrification (HiEl): This scenario increases the stringency of the $CO_2$ price from \$50/t-$CO_2$ to \$100/t-$CO_2$ in 2025, escalating at 10% per year (instead of 7% in the reference). This scenario also assumes additional technology and policy drivers that accelerate electrification by lowering the cost of end-use technologies, reducing customers' reticence to shift technologies, and accelerating the turnover of equipment. Accelerated cost reductions in this scenario could be interpreted either as faster-than-expected cost declines or as policy-driven incentives.

Load management is incorporated through deferrable electric vehicle charging in both scenarios.

## Caveats

There are several limitations to keep in mind when interpreting the analysis. First, the analysis assumes that fossil fuel price trajectories over time are exogenously specified. Second, the analysis is not explicitly modeling operational constraints (e.g., inertia) or detailed ancillary services markets. Third, hydrogen is considered only for new investments and not for blending in or retrofits of existing units. Fourth, modeling of investments and operations is conducted at an hourly level, and there is no subhourly or sub-state detail (e.g., transmission and distribution constraints), though costs are included. Finally, the analysis uses a single historical weather year (2015) for consistency of meteorologically driven time series data.

Additional model details and input assumptions are provided in EPRI (2020) and Supplementary Information (Supplementary Notes 1 and 2).

## Data availability

The optimization data that support the analysis within this paper are available from the repository https://github.com/b3311/gasnetzero.

## Code availability

The optimization code that support the analysis within this paper is available from the repository https://github.com/b3311/gasnetzero.

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

## Acknowledgements

The authors would like to thank Geoff Blanford and Adam Diamant for their helpful feedback. Thanks to Naga Srujana Goteti for research assistance. The views expressed in this paper are those of the individual authors and do not necessarily reflect those of EPRI or its members.

## Author contributions

J.E.T.B and D.T.Y. developed the model, conceived the study, designed scenarios, and contributed to writing the article. J.E.T.B. led the modeling, data analysis, and article revisions.

## Competing interests

The authors declare no competing interests.
