## [Peer Review File · Nature Communications]

Reviewer comments, first round

Reviewers' comments:

Reviewer #1 (Remarks to the Author):

Is There a Role for Natural Gas in Reaching Net-Zero Emissions in the Electric Sector?
Peer Review, February 2022

I appreciate the opportunity to review this manuscript and thank the authors for their clear and concise contribution. The topic of GHG-emitting technologies in the context of grid decarbonization is a very important part of climate policy and economics and I am grateful to see the authors' efforts to deliver well-structured analysis here. Their results demonstrate that, when coupled with negative emissions technologies such as carbon capture and sequestration and direct air capture, natural gas will continue to play a significant role in a least-cost US electric grid for at least the next 15 years under a wide range of technology, policy, and market assumptions. This finding is welcome amidst the scale of change required to meet net-zero targets and following substantial investment in natural gas capacity over the past 15 years. The authors provide a unique contribution in the level of techno-economic complexity and temporal resolution the REGEN model offers the question of natural gas generation in a (net) zero GHG emissions 2035.

This manuscript can make a valuable contribution to the literature with some revision. Many prior studies have questioned the significance of the role for additional natural gas capacity but have stopped short of arguing that natural gas will play no role all together. It is an understandable challenge to distill the volume of insight REGEN delivers on this topic into an article of this length and format. Still, the manuscript can contribute more than establishing the existence of a role for natural gas by further quantifying and qualifying that role. I provide thematic comments immediately below that should be addressed for publication. I then provide conceptual and clarifying comments for the authors' and editor's further consideration.

Thematic

- Capacity additions: capacity additions can help qualify whether there is any additional role for natural gas to play or if current capacity is sufficient. Include discussion of natural gas capacity additions relative to current installed capacity with respect to the following points:
 - o Figure 22 indicates natural gas generation is declining in the stringency of the target. Please discuss the extent natural gas capacity additions (inclusive of new and retrofit CCS) in the context of this decline in generation.
 - o Figure 13 appears to suggest no new gas capacity (i.e., the investment bar) post 2025. Does this indicate a "break-even" year for natural gas capacity?
 - o Recommend including bars for 2020 and 2035 reference results at the left of Fig 1A, 1B.
- Asset values: changes in the value of existing gas capacity can help quantify the extent to which gas capacity plays a positive role in meeting targets. Expand on the discussion of the return to natural gas generating capital in the net-zero case [173-176] relative to the reference case, which will indicate asset impairment. Please state your model assumptions on economic retirement.
- Generation costs: The authors' discussion of generation cost is a helpful quantification of natural gas capacity's role. Given the criticality of negative emissions technologies in meeting the target with gas, please state the assumed cost of DAC, the extent of its use, and whether it is reflected in your electricity prices. Recommend the cost of DAC be included in your wholesale prices in as much as it is required to achieve the target. Note that a net-zero policy might require GHG-emitting generators to purchase offsets, which would raise the cost of gas generation without CCS.
- Results comparisons: The significance of Figure 5 is very difficult to interpret because the targets are not stated in the other models. For example, the carbon tax scenarios do not achieve the same decarbonization goals. Capacity might also be masking significant under-utilization. Recommend striking this figure or making scenario-consistent comparisons only.

Conceptual

- The 1.5% leakage assumption looks like a reference case assumption. Can the author's motivate its exclusion, if only in a footnote, as consistent with the NDC? Consider including the 1.5% leakage in the reference case or moving this case to the supplemental – the value of presenting

both 1.5% and 3.0% was not clear.

- Can you provide an interpretation for the appearance of new nuclear in the 3% leak scenario but in no others? For example, the NoGas and NoGasCCS scenarios also raise
- [115-6]: "higher capital cost and higher capacity factor option but is only deployed up to a point versus wind and solar, after which uncaptured gas of some form (either NGCC or peaking units) is preferred." Explain the significance of this statement; clarify at what "point," and why?
- Does the identical investment costs for net zero and carbon free 2050 imply no system value to natural gas in that period (Figure 2)? How do the cumulative investment costs through 2050 compare? How much new gas is built?
- How is the policy implemented in the model and where is the rent on the carbon constraint accounted?

Clarifying

- [54]: Note that "gas has a role in the least-cost path ... except when policy design precludes it" but in intro say it requires availability of NETs. Would abstract be more consistent to read "... where negative emissions technologies are allowed under policy"?
- [57; and throughout]: define the capacity qualifier "firm" at first use.
- [123]: the authors say that CDR is from BECCS, which enables gas emissions. But there are scenarios with NGCC generation and no BECCS. Is DAC enabled in all scenarios or just NoGasCCS as discussed on [128]?
- [183]: the authors say the South region has relatively less solar resource capacity. This makes sense relative to the West but is counter intuitive with respect to the East. Is there a simple way to quantify resource capacities to make this clearer?
- [203]: the authors critique identification of "stranded assets" for not fully considering carbon removal technologies. Recommend a more nuanced consideration of asset impairment in lieu of stranding (i.e., complete impairment). (See asset value comment above.)
- [Figures 4&5]: is there any meaning to the horizontal sides of dots within a year?
- [Figure 1B]: suggest legend entry for average generation price and generation price would make more sense in Figure 1A (generation)
- [261] has a reference error.

Sincerely,
Jared Woollacott

Reviewer #2 (Remarks to the Author):

Review of "Is There a Role for Natural Gas in Reaching Net-Zero Emissions in the Electric Sector?" by Bistline and Young, submitted to Nature Communications. Reviewed February 2022.

- Will the work be of significance to the field and related fields? How does it compare to the established literature? If the work is not original, please provide relevant references.

The topic of what generating portfolios will allow the U.S. to reach net-zero GHG emissions from electricity in 2035 is important, since it is a component of the nationally defined contribution submitted by the U.S. government to the UNFCCC. As far as I can tell from a brief search, the authors' claim that this would be the first scholarly publication on that specific topic is true.

The role of natural gas generation in reaching net zero, whether in 2035 or in 2050, is also important. It is addressed in this study by contrasting "net-zero" vs "carbon-free" scenarios, with the former having significant uncaptured natural gas generation offset by negative emissions, and the latter having natural gas generation with 100% carbon capture. The question of what kinds of generation to promote to move the electricity industry toward zero emissions is a subject of policy debate at both the state and federal level in the U.S. Relatedly, there are current policy initiatives in some states to prevent new natural gas generation from being built and/or to phase out all natural gas generation. This paper informs these policy debates.

- What are the noteworthy results?

The general finding that natural gas generating capacity can reduce costs of net-zero scenarios in 2035 relative to carbon-free scenarios (Figure 2) is an important finding and consistent with other recent analyses focused on net-zero in 2050. The levels of gas capacity required in 2050 is also broadly consistent with these studies. The finding that capacity factors for the natural gas generating fleet will decline over time (Figure 16) is also consistent with these analyses. The discussion of the effect of changing capacity factors on the economics of gas generators is of interest (line 521 et. seq.).

On the other hand, compared to recent net-zero studies for 2050, this study finds much higher levels of gas generation and much lower levels of solar generation and variable renewable generation overall. Only the “pessimistic gas” scenario (Figure 1A) shows a renewable generation share comparable to that in other studies, and that in cases where the renewable buildout was constrained due to siting or other limitations (i.e. the most pessimistic scenarios for renewables, and the most optimistic for gas). Technology and fuel cost assumptions are similar across this study and other recent studies, so the differences in results lie elsewhere.

- Is there enough detail provided in the methods for the work to be reproduced? Does the work support the conclusions and claims, or is additional evidence needed? Are there any flaws in the data analysis, interpretation and conclusions? Do these prohibit publication or require revision?

A major concern about this study is that the boundary conditions for the electricity sector are poorly specified. The authors refer in passing to electrification loads related to economy-wide decarbonization in a qualitative way (e.g. line 105), but the composition of load in 2035 and 2050 is not specified in the text or in figures. Perhaps even more important, economy-wide emissions targets are not specified, and emissions figures are not provided, so the amount of decarbonized electricity required for new loads economy-wide can't be easily deduced.

This lack of clarity about demand highlights the anomalous result of Figure 12 (also Figures 1A, 18, and 20) which shows 2035 generation across all scenarios being very similar to today's, and to the presumably low-electrification 2035 reference case. This is not what would be expected in an economy undergoing rapid electrification of end uses in transportation, buildings, production of hydrogen for fuel, etc. Figure 18 shows generation in 2050 that is roughly half the level found in recent net-zero studies. Even though this study is framed as electricity-sector only, it needs at a minimum to explore different levels of demand, since the rest of the economy is an unavoidable boundary condition for the electricity sector and demand would vary widely with economy-wide policies. Almost all of the relevant metrics for the sector – emissions intensity, capacity mix, generation mix, balancing requirements, costs, rate impacts, and resource use – depend on the level of demand and its composition. Perplexingly, Figure 17 shows the U.S. being a winter-peaking system in 2035 for some reason. This is a major shift from the historical U.S.-wide summer peak. Winter-peaking could be consistent with a high building electrification load, but if this is the case, why is there only a minimal increase in annual demand?

The scale of demand is a key driver of the negative emissions and carbon capture required in the net zero scenarios, including how much biomass fuel for BECCS is required to offset emissions from uncaptured natural gas generation. Despite the key role of BECCS in reaching net zero, the carbon accounting is not shown, nor are the annual biomass requirements for generation fuel, and it is not discussed in the paper how this would compare to sustainable levels of biomass use. The conclusions of this study could look very different if higher electricity demand caused the resulting portfolios to bump up against the limits of carbon capture and negative emissions, due to supply curve steepness or biomass constraints or geologic sequestration limits.

This concern is reinforced by Figure 23, which shows high levels of natural gas use remaining in the overall U.S. economy in 2050. It seems inconsistent to assume U.S. policies that would require zero emissions in electricity while allowing major emissions from natural gas use to continue in the rest of the economy. After all, one of the main motivations driving electricity decarbonization policies is the use of decarbonized electricity to displace fossil fuel use elsewhere in the economy. One concludes either that there is incipient electricity demand that was not incorporated in the authors' analysis due to limiting the scope of electrification of end uses and the production of

decarbonized fuels, or that there is an implicit assumption of very high economy-wide carbon capture and storage or direct air capture (but if so this is also not evident in electricity demand). Note that residual petroleum use and its associated emissions is not addressed, but would further stretch the limits of negative emissions and biomass use.

A second concern is a lack of clarity about the modeling and how the modeling might effect the results. A simple example is that the text (line 441 et. seq.) says that the hourly dispatch was done in the static mode, and that the static mode was used only for the years 2035 and 2050. If that is the case, how were the gas capacity factors for every 5 years from 2025 to 2050 obtained? A more important question not answered in the methodology descriptions is whether the US-REGEN model has the ability to represent sector coupling between the electricity and fuels production sectors. If not, it is difficult for the authors model to accurately represent effects that have a strong influence on the central tradeoffs being explored in this paper, for example battery storage vs gas capacity, the cost of producing hydrogen by electrolysis, etc. In this regard, the modeling may not have the capacity to give confidence regarding some of the claims of the paper, and the authors should provide stronger caveats. Even without knowing the intimate details of the model, there is evidence to support this concern. The low values for solar throughout indicate that load flexibility is limited in the modeling –the ability to do flexible charging or to shift to large industrial flexible loads like electrolysis seems to be low; if not, there would be more solar and more hydrogen production. (It appears that “green hydrogen” in the fuel mix is exogenous, not an endogenous result of sector coupling). This could be due to the apparent low level of electrification mentioned earlier, or to inability to model sector coupling. One speculates that the latter might be because the US-REGEN model does not have the ability to represent cumulative energy surpluses and deficits (aka state of charge) across all hours of the year (see e.g. “Optimal gas-electric energy system decarbonization planning” by Von Wald et al 2022, starting on p.4, for a recent treatment of this question), which is necessary to represent sector-coupling accurately.

Final observations. (1) Several of the results figures, for example Figures 4, 5 and 15, show such wide ranges that they are not really informative and one can draw few definitive conclusions from them. This problem is not confined to the present manuscript, but is found too widely in energy modeling studies. It is nonetheless not of the standard that would be expected in a Nature journal. (2) The generic use of “gas” as a substitute for “natural gas” for brevity (line 26) is problematic. The authors themselves do not adhere strictly to this convention, and this usage is confusing in situations in which there could be expected to be gas blends with varying levels of fossil natural gas (3) Some of the writing in the paper is less clear than it could be. An example is the last sentence of the abstract. (4) There is an “error reference not found” at line 261.

- Overall recommendation

It is difficult to recommend publication of this manuscript in a Nature journal. The fact that the top line results support findings that have been found in other recent studies regarding the value of gas generating capacity in maintaining reliability in a net zero electricity system does not redeem the paper’s other flaws. It is not a novel result to conclude that inexpensive natural gas coupled with seemingly unlimited negative emissions capacity will favor a net-zero scenario over a carbon-free scenario in an optimization model. If the paper were returned to the authors for a rewrite, their main task would be to address the lack of clarity about boundary conditions regarding emissions and electricity demand through new model runs, new scenarios, and a thorough re-write, with much greater detail about what is assumed for the rest of the economy, including economy wide energy mixes and emissions, including biomass use, negative emissions, and CCS.

Reviewer #3 (Remarks to the Author):

Key results

The authors analyze the role of natural gas in combination with carbon capture, sequestration and

removal technologies in decarbonization scenarios for the United States. To do so, they use an optimization model that has been developed by EPRI and that has already been applied in previous research. Their main finding is that natural gas plays a substantial role in all investigated decarbonization scenarios under the assumptions made here. The only exception is a "pessimistic" sensitivity where several assumptions are combined that make power generation from natural gas unfavorable. The authors also highlight that wind and solar have higher generation shares than natural gas in all scenarios analyzed.

Validity

In general, I think the main part of the analysis and the major conclusions drawn are not so much about natural gas (or its "system value") per se, but rather about gas in combination with CCS technologies, including carbon removal via BECCS or DAC. So an alternative (and maybe more appropriate) title of the paper may be something like "how carbon capture and removal can help in achieving net zero emissions in the electric sector". In the Discussion, the authors state "Our results indicate that new and existing gas can help to reduce emissions..." – but in fact, it is predominantly CCS/BECCS that facilitates net-zero emissions here, and not so much natural gas as such.

Yet, I find it hard to fully grasp the role of natural gas in combination with CCS/DAC technologies, as there is no alternative scenario in which BECCS/DAC are fully excluded (also not in "NoGasCCS"). In such a setting, alternative decarbonization strategies such as combinations of renewables and storage would have to be used. There is only the "pessimistic" scenario, but here, several sensitivities are combined together, and CCS is still not fully excluded. At least from a European perspective (which I have), the assumption that no BECCS/DAC may become available for the power sector appears to be policy relevant. In fact, several European countries currently struggle with the development of CCS infrastructure to deal with hard-to-abate industrial / process emissions – let alone develop additional CO₂ transportation and storage infrastructure for CO₂ from gas-fired power plants. This may be different in the US – but still, the effect of gas+CCS could be better quantified against a reference without any CCS. So in my view it would strengthen the validity and robustness of results if there was a setting without any CCS/BECCS/DAC technologies. This would also support the statement "Natural gas' role in a net-zero system hinges on carbon capture..." (line 208).

A related point: I guess that the cost assumptions for CCS/BECCS/DAC technologies are relatively favorable compared to renewable costs, judging from Figure 8 and from the relatively low optimal renewable shares in decarbonization scenarios. Yet, DAC and carbon storage costs are not shown in Figure 8, and they are also not provided elsewhere as far as I can see (maybe I missed it).

Further, I think the parameterization of scenarios and sensitivities could be revisited. For example, I don't see why the reference assumes zero upstream methane leakage – this is clearly unrealistic. I guess the "1.5% Leak" sensitivity could be the baseline assumption. Likewise, it is not clear if some long-term emissions from captured/sequestered carbon are accounted for in the analysis. I am not an expert in this field, but the issue of long-term CO₂ leakage from storage/sequestration could at least be discussed (or represented in the parameterization, if relevant).

Next, there are several statements in the manuscript which appear somewhat bold and/or under-developed to me. For example, the authors claim that natural gas is relevant "not only during the transition to net-zero emissions but also at the destination" (lines 100-101). Yet, hardly any results are provided for the transition part of this statement, the focus is clearly on the 2035 "destination". Likewise, the statement "The extent of gas generation for a decarbonized electric sector depends on policy design, ability to mitigate upstream methane, and transition risks from technological change" (107-108) is extremely broad and not fully connected to the results presented. Another example: "...least-cost decarbonization pathways include emerging technologies such as advanced nuclear, hydrogen, and capacity equipped with carbon capture and sequestration" (110-112): this is a very general statement that appears to be not really supported by the results presented. CCS in fact is always a relevant part of the solution in the results presented (except for "Pess"), but advanced nuclear and hydrogen are relevant only in very specific cases. To give another example, I find it hard to relate the statement made on line 136

"Accelerating decarbonization entails greater contributions from gas on a relative and absolute basis..." to the material presented in Figure 4.

Finally, the authors further seem to suggest that decarbonization gets cheaper if we wait longer, as renewable energy costs will by assumption decrease over time (cp. Lines 159 ff.). I think this may lead to potentially flawed conclusions. This conclusion is clearly driven by the assumption that renewables get cheaper over time, irrespective of the speed of deployment (in the US and globally). Further, the IPCC has repeatedly stated that the costs of climate change increase, the longer we wait with mitigation. Maybe the authors don't mean to suggest this, but the reader may interpret these statements such that delaying decarbonization ambitions would be preferable.

Significance

In general, I think the research question is timely and policy relevant.

Yet, at least from an EU perspective, the reader probably would expect some other insights from a paper with this title. In Europe, we currently have a fierce debate about the role of natural gas for the decarbonization of the energy system. This recently became very visible in the context of the taxonomy debate, i.e. can investments in natural gas power plants be considered sustainable or not. An important question here is not only for how long natural gas can be used for generating electricity, but also how and when it could be substituted partly or completely by (low-carbon) hydrogen. By design, the analysis is completely silent on this aspect, as it allows the use of natural gas also in the future via CCS/BECCS/DAC.

The finding that renewables have higher generation shares than natural gas in cost-optimized decarbonization scenarios is not very surprising and also not exactly new. In fact, this may be considered a well-established fact that has been demonstrated in various decarbonization studies in the literature.

Data and methodology, analytical approach

The authors provide a high-level description of the model they use, and refer to a more detailed model description published by EPRI. The model itself is not provided open-source as far as I can see (or did I miss it?). Especially when it comes to the concrete input parameters used, this leads to a lack of transparency, and it impedes reproducibility. I think it is fair to consider the open-source provision of models and input data as best practice in the energy modeling field, and I personally would also expect this from research published in a highly ranked journal such as *Nature Communications*.

The renewable cost assumptions appear to be rather pessimistic compared to previous analysis, especially for solar PV. For example, compare the analysis by Victoria et al. published in the same journal (<https://doi.org/10.1038/s41467-020-20015-4>), where renewable costs are much lower. Looking at the numbers provided there, the "LoRE" sensitivity could probably be considered as the baseline here. In turn, the CCS / BECCS / DAC cost assumptions are probably relatively optimistic (but they are not fully transparent, see remark above). Are there variable costs of bioenergy? In fully decarbonized energy systems, it is quite plausible that bioenergy is scarce (and thus, expensive), as it is likely to be used in the transportation sector (and maybe for heating). Another question is if the additional load required for DAC is accounted for in the model. At least this is not visible in Figure 1.

Concerning the methodology, I did not fully understand to what extent the model is dynamic. Of course, a dynamic model would be very desirable for this kind of analysis, and would distinguish this analysis from several previous papers. Yet, the Supplemental Information suggests that the dynamic mode is used only for retirement of existing capacity. But again, this is hard to check as model code is not provided.

Likewise, it became not clear to me if capacity decisions related to both energy and power are endogenous only for batteries, as suggested in the respective paragraph in the SI, or also for other storage technologies.

Regarding the electricity prices shown in Figure 3: Here the reference has the lowest prices, and decarbonization scenarios are always more expensive. When prices are interpreted as an indicator of "costs", which seems to be the case here, this may lead to flawed conclusions, as there are substantial external costs related to non-abated carbon emissions in the reference. If I understand correctly, CO₂ is not priced in the reference.

Suggested improvements

I think the scenario space of this study could be reduced. The authors introduce three policy targets (reference, carbon-free and net-zero), as well as two target years (2035 and 2050). Yet, most of the presented results focus on the net-zero 2035 setting. So it would be clearer and less confusing to make this focus more clear already in the beginning of the results section.

I generally like the idea of comparing the results to other studies, but the respective paragraph in the Discussion section (including Figure 5) is underdeveloped in my opinion. For readers not very familiar with the other studies mentioned, it is very hard to understand Figure 5 and the different results provided for 2030 (?) and 2050.

A list of selected other points where I see some potential for improvements:

- Line 27: the statement seems to be a fairly general one, but both references only refer to the U.S.
- Line 38 ff: The contributions may be better placed after the following paragraph where the research question is stated
- Is it plausible that an increase in the non-electric gas use would be allowed in the carbon-free scenario (lines 157ff.)?
- The "Regional Differences in Decarbonization Strategies" section appears to be quite US-specific. It is not clear to me if the general findings could also be transferred to other world regions.
- Figure 4 looks nice, but I find it hard to understand. Especially, the differences between 2035 and 2050 are hard to see. I am also not sure how to interpret results because of the aggregation ("Shares include natural-gas-193 fired generators with and without carbon capture as well as hydrogen through steam methane reforming").
- In the Discussion section, it is argued that natural gas would be compatible with net-zero goals if "(2) the electricity market structure effectively values capacity in high-renewables systems". I am not sure that this can be considered a direct outcome of the study; it rather seems to be an implicit conclusion (as capacity is actually valued in the optimization model).
- Onshore / offshore wind are different technologies with different costs, but results are only shown for "wind" (maybe no offshore in the optimum?)
- Very recently, the Russian attack on the Ukraine has intensified the question how to substitute natural gas in the European power sector. The analysis could probably connect to this development and discuss the implications.

References

The references appear to be somewhat unbalanced in my opinion. The manuscript includes 38 references, of which one is mentioned twice, so 37 unique references overall. Of these, 13 are grey literature, and 24 are peer-reviewed. The peer-reviewed articles are heavily focusing on the US. I think it would be reasonable to also connect to related model analysis from other world regions. In particular, many relevant model-based decarbonization studies have been published for Europe, which should not be ignored here. What is more, of the 24 peer-reviewed references, 12 are self-citations. I fully appreciate that the authors have made important contributions to the field before – but I think this is an excessive level of self-citations.

Response to Reviewers

Reviewer #1

Comment 1.1: “I appreciate the opportunity to review this manuscript and thank the authors for their clear and concise contribution. The topic of GHG-emitting technologies in the context of grid decarbonization is a very important part of climate policy and economics and I am grateful to see the authors efforts to deliver well-structured analysis here. Their results demonstrate that, when coupled with negative emissions technologies such as carbon capture and sequestration and direct air capture, natural gas will continue to play a significant role in a least-cost US electric grid for at least the next 15 years under a wide range of technology, policy, and market assumptions. This finding is welcome amidst the scale of change required to meet net-zero targets and following substantial investment in natural gas capacity over the past 15 years. The authors provide a unique contribution in the level of techno-economic complexity and temporal resolution the REGEN model offers the question of natural gas generation in a (net) zero GHG emissions 2035.

This manuscript can make a valuable contribution to the literature with some revision. Many prior studies have questioned the significance of the role for additional natural gas capacity but have stopped short of arguing that natural gas will play no role all together. It is an understandable challenge to distill the volume of insight REGEN delivers on this topic into an article of this length and format. Still, the manuscript can contribute more than establishing the existence of a role for natural gas by further quantifying and qualifying that role. I provide thematic comments immediately below that should be addressed for publication. I then provide conceptual and clarifying comments for the authors’ and editor’s further consideration.”

Response 1.1: We thank the reviewer their assessment of the importance of the issues addressed in our paper and of the contributions of our work.

*Comment 1.2: “Capacity additions: capacity additions can help qualify whether there is any additional role for natural gas to play or if current capacity is sufficient. Include discussion of natural gas capacity additions relative to current installed capacity with respect to the following points:
o Figure 22 indicates natural gas generation is declining in the stringency of the target. Please discuss the extent natural gas capacity additions (inclusive of new and retrofit CCS) in the context of this decline in generation.”*

Response 1.2: We added Supplementary Figure 20 in SI Note S3 to show installed capacity across scenarios and to compare this capacity with the existing fleet. We also added explanatory text: “Installed natural gas capacity across scenarios is shown in Figure 20. Relative to the current fleet in 2020, many of these scenarios represent a slight increase on net or modest decreases. However, the composition of the natural gas fleet can differ across scenarios, as retirements of existing capacity are replaced by CCS-equipped units or hydrogen turbines.”

Comment 1.3: “Figure 13 appears to suggest no new gas capacity (i.e., the investment bar) post 2025. Does this indicate a “break-even” year for natural gas capacity?”

Response 1.3: We added text in the paragraph before Figure 13 to convey that the values over time represent cash flows for a hypothetical NGCC plant: “To illustrate the economics of natural gas additions with deeper CO₂ reductions, Supplementary Figure 13 shows revenues and costs over time for a hypothetical NGCC plant built in 2025 under the Net-Zero by 2035 scenario. Upfront investment costs are shown for the year the plant comes online (2025). Net operating revenue is relatively flat over time due to the firm back-up role for variable renewables, as gas typically sets the price in energy markets. A

large fraction of revenues (roughly 70% for this scenario and region) occur in the first 15 years of operation.” The caption was also reworded: “Revenues and costs for an illustrative NGCC plant over time in the Net-Zero by 2035 scenario. All cash flows are shown in discounted terms for a 2025 vintage NGCC plant in the Mid-Atlantic model region.”

Comment 1.4: “Recommend including bars for 2020 and 2035 reference results at the left of Fig 1A, 1B.”

Response 1.4: We added bars for 2020 and the 2035 reference in Figure 1.

Comment 1.5: “Asset values: changes in the value of existing gas capacity can help quantify the extent to which gas capacity plays a positive role in meeting targets. Expand on the discussion of the return to natural gas generating capital in the net-zero case [173-176] relative to the reference case, which will indicate asset impairment. Please state your model assumptions on economic retirement.”

Response 1.5: We added a SI figure and text to illustrate how energy and capacity returns vary between the reference and policy scenarios: “Supplementary Figure 25 shows normalized revenues for existing NGCC capacity, which are proxies for the market value of capacity. Energy revenues refer to bulk electricity sales, and capacity revenues relate to the ability to provide firm capacity contributions during periods of system stress. Under reference policy conditions, energy revenues are the dominant value stream for NGCC plants in many regions. However, with the Net-Zero by 2035 policy, revenues shift from energy to capacity, but overall returns are much lower with more stringent policy. There is considerable regional heterogeneity in the magnitude of this asset impairment, which varies based on regional natural gas prices, the existing asset mix, renewable resource characteristics, and extent of existing plant retirements. Note that the model endogenously retires existing capacity when the net present value of going-forward costs exceeds that of projected revenues in a given scenario.”

We also added sentences after Figure 3 in the main text to summarize: “Moving from a reference to a net-zero policy environment lowers returns to existing NGCC capacity and shifts its value from bulk electricity sales to firm capacity (Supplementary Figure 25). The magnitude of this asset impairment varies considerably by region.”

Comment 1.6: “Generation costs: The authors discussion of generation cost is a helpful quantification of natural gas capacity’s role. Given the criticality of negative emissions technologies in meeting the target with gas, please state the assumed cost of DAC, the extent of its use, and whether it is reflected in your electricity prices. Recommend the cost of DAC be included in your wholesale prices in as much as it is required to achieve the target. Note that a net-zero policy might require GHG-emitting generators to purchase offsets, which would raise the cost of gas generation without CCS.”

Response 1.6: We added text, figure, and table to Supplemental Information Note S2 to describe the representation of DAC, parametrization, and data sources. We modified the caption for Figure 3 to reflect that costs of DAC are included: “U.S. generation-weighted average generation price (\$/MWh), which reflects all generation, new bulk transmission, carbon removal, and CO₂ transport and storage costs.” In the paragraph after Figure 1, we added a sentence to note that, “A Net-Zero policy might require emitting resources to purchase CDR credits, which would raise the dispatch costs of natural-gas-fired generators.”

Comment 1.7: “Results comparisons: The significance of Figure 5 is very difficult to interpret because the targets are not stated in the other models. For example, the carbon tax scenarios do not achieve the same decarbonization goals. Capacity might also be masking significant under-utilization. Recommend striking this figure or making scenario-consistent comparisons only.”

Response 1.7: We agree with the reviewer's points and have removed Figure 5 from the Discussion section, instead opting to make qualitative comparisons with other studies in the text.

Comment 1.8: "The 1.5% leakage assumption looks like a reference case assumption. Can the author's motivate its exclusion, if only in a footnote, as consistent with the NDC? Consider including the 1.5% leakage in the reference case or moving this case to the supplemental – the value of presenting both 1.5% and 3.0% was not clear."

Response 1.8: We reran all of the scenarios with a 1.5% upstream CH₄ leakage rate as the default assumption with 3% as the sensitivity. We added text to the "Methods" section on this reference assumption: "The emissions intensity of gas includes upstream gas system CH₄ releases with a 1.5% leakage rate (i.e., volumetric fraction of consumed natural gas). This 1.5% rate is similar to the EPA's Inventory of U.S. GHG Emissions and Sinks and bottom 95% of sites from a recent study [31]. A 100-year Global Warming Potential of 25 is used based on the UNFCCC standard used by the U.S. EPA." The scenario description in Table 1 mentions that: "Adjust upstream gas system CH₄ releases with 3% leakage rate (instead of the reference assumption of 1.5%)."

Comment 1.9: "Can you provide an interpretation for the appearance of new nuclear in the 3% leak scenario but in no others? For example, the NoGas and NoGasCCS scenarios also raise."

Response 1.9: With the updated leakage rate and addition of the Carbon-Free scenario to Figure 1, there are now new nuclear additions in the NoGasCCS, 3% Leak, and Carbon-Free scenarios. The interpretation is that, as the choice set of firm technologies is constrained (NoGasCCS) or made more costly (3% Leak), the portfolio moves closer to the Carbon-Free scenario where carbon removal and natural gas are prohibited. We added text after Figure 1: "As the choice set of firm technologies is constrained or made more costly, the generation and capacity mixes move closer to the "Carbon-Free" (CF) scenario where carbon removal and natural gas are prohibited."

Comment 1.10: "[115-6]: "higher capital cost and higher capacity factor option but is only deployed up to a point versus wind and solar, after which uncaptured gas of some form (either NGCC or peaking units) is preferred." Explain the significance of this statement; clarify at what "point," and why?"

Response 1.10: We expanded this sentence and added a reference to Supplementary Figure 22, which shows capacity factors by capacity type: "NGCC with CCS plays a different role than uncaptured gas: The former is a higher capital cost and higher capacity factor option, which is preferred to new nuclear but only deployed when it can operate with capacity factors above roughly 60% to balance wind and solar, after which uncaptured gas of some form is preferred."

Comment 1.11: "Does the identical investment costs for net zero and carbon free 2050 imply no system value to natural gas in that period (Figure 2)? How do the cumulative investment costs through 2050 compare? How much new gas is built?"

Response 1.11: The very similar investments and costs through 2035 across the two policy variants suggest that the policy definition does not have as large of an impact on near-term investments. The transition path for both policy variants starts by adding a lot of wind and solar, and a smaller amount of battery storage, plus retiring coal. Differences in technology pathways appear beyond 80% CO₂ reductions. We added Figure 6 to highlight differences in 2035 and 2050.

Comment 1.12: "How is the policy implemented in the model and where is the rent on the carbon constraint accounted?"

Response 1.12: We added a clause to “Modeling Deep Decarbonization in the Electric Sector” that the emissions policy is “...implemented as a constraint on national CO₂ emissions corresponding to the emissions trajectories in Supplementary Figure 8 in Note S2.” We added a paragraph on the shadow price on the CO₂ constraint to the “System Value of Natural Gas” section: “Policy stringency and the value of natural gas and carbon removal also can be evaluated by comparing shadow prices on the CO₂ emissions cap constraint. Marginal abatement costs increase sharply as zero emissions without CDR are approached, which are approximately \$90,000/t-CO₂ by 2035 in the Carbon-Free scenario. With CDR in the Net-Zero scenario, marginal abatement costs corresponding to the scenarios in Figure 1 range from \$77/t-CO₂ (LoRE) to \$126/t-CO₂ (Pess), which is the marginal cost of capture from BECCS.”

Comment 1.13: “[54]: Note that “gas has a role in the least-cost path ... except when policy design precludes it” but in intro say it requires availability of NETs. Would abstract be more consistent to read ‘.. where negative emissions technologies are allowed under policy’?”

Response 1.13: We altered this sentence in the abstract to read: “We find that gas-fired generation can lower the cost of electric sector decarbonization, a result that is robust to a wide range of sensitivities, where negative emissions technologies are allowed under policy.”

Comment 1.14: “[57; and throughout]: define the capacity qualifier “firm” at first use.”

Response 1.14: We added a footnote upon the first use of “firm” in the Results section: “Here, clean firm resources refer to “technologies that can be counted on to meet demand when needed in all seasons and over long durations” (Sepulveda, et al., 2018) such as hydro, biomass, geothermal carbon-capture-equipped capacity, and zero-carbon gas-fueled plants (e.g., hydrogen).”

Comment 1.15: “[123]: the authors say that CDR is from BECCS, which enables gas emissions. But there are scenarios with NGCC generation and no BECCS. Is DAC enabled in all scenarios or just NoGasCCS as discussed on [128]?”

Response 1.15: We reported DAC capacity across scenarios in Figure 1B. We added text after Figure 1 that, “BECCS is the main CDR technology used to reach Net-Zero goals due to its lower cost of net CO₂ removal and provision of firm negative-CO₂ electricity; however, sensitivities in Note S3 show how DAC deployment can increase under alternate technological cost and availability assumptions.”

Comment 1.16: “[183]: the authors say the South region has relatively less solar resource capacity. This makes sense relative to the West but is counter intuitive with respect to the East. Is there a simple way to quantify resource capacities to make this clearer?”

Response 1.16: We revised Figure 3 to show generation by region and electricity prices across the electric sector policy scenarios, which should clear up confusion about the relative roles of wind and solar across different regions.

Comment 1.17: “[203]: the authors critique identification of “stranded assets” for not fully considering carbon removal technologies. Recommend a more nuanced consideration of asset impairment in lieu of stranding (i.e., complete impairment). (See asset value comment above.)”

Response 1.17: We added a sentence to this passage in the Discussion section to point out these more nuanced dynamics of asset impairment: “Such binary framings also often overlook more nuanced conditions of asset impairment such as when returns to existing capital are lower but not necessarily low enough to induce retirement (Supplementary Figure 25).”

Comment 1.18: “[Figures 4&5]: is there any meaning to the horizontal sides of dots within a year?”

Response 1.18: We added a sentence to the caption of Figure 4 that, “Note that points are jittered horizontally within a column for greater visibility of individual values.”

Comment 1.19: “[Figure 1B]: suggest legend entry for average generation price and generation price would make more sense in Figure 1A (generation)”

Response 1.19: We altered this figure so that Figure 1A includes the generation price on the secondary axis and that Figure 1B includes DAC capacity on the secondary axis. Legend entries were added.

Comment 1.20: “[261] has a reference error.”

Response 1.20: We fixed this reference error.

Reviewer #2

Comment 2.1: “The topic of what generating portfolios will allow the U.S. to reach net-zero GHG emissions from electricity in 2035 is important, since it is a component of the nationally defined contribution submitted by the U.S. government to the UNFCCC. As far as I can tell from a brief search, the authors’ claim that this would be the first scholarly publication on that specific topic is true.

The role of natural gas generation in reaching net zero, whether in 2035 or in 2050, is also important. It is addressed in this study by contrasting “net-zero” vs “carbon-free” scenarios, with the former having significant uncaptured natural gas generation offset by negative emissions, and the latter having natural gas generation with 100% carbon capture. The question of what kinds of generation to promote to move the electricity industry toward zero emissions is a subject of policy debate at both the state and federal level in the U.S. Relatedly, there are current policy initiatives in some states to prevent new natural gas generation from being built and/or to phase out all natural gas generation. This paper informs these policy debates.”

Response 2.1: Thank you for your encouraging comments and constructive feedback, which have been helpful as we refined the manuscript.

Comment 2.2: “The general finding that natural gas generating capacity can reduce costs of net-zero scenarios in 2035 relative to carbon-free scenarios (Figure 2) is an important finding and consistent with other recent analyses focused on net-zero in 2050. The levels of gas capacity required in 2050 is also broadly consistent with these studies. The finding that capacity factors for the natural gas generating fleet will decline over time (Figure 16) is also consistent with these analyses. The discussion of the effect of changing capacity factors on the economics of gas generators is of interest (line 521 et. seq.).

On the other hand, compared to recent net-zero studies for 2050, this study finds much higher levels of gas generation and much lower levels of solar generation and variable renewable generation overall. Only the “pessimistic gas” scenario (Figure 1A) shows a renewable generation share comparable to that in other studies, and that in cases where the renewable buildout was constrained due to siting or other limitations (i.e. the most pessimistic scenarios for renewables, and the most optimistic for gas). Technology and fuel cost assumptions are similar across this study and other recent studies, so the differences in results lie elsewhere.”

Response 2.2: We updated the analysis with lower variable renewables costs and an upstream CH₄ leakage rate of 1.5% for natural gas in all scenarios. These changes lead to higher wind and solar generation shares in Figure 1. As the reviewer suggests, technology and fuel cost assumptions align with other studies in the literature, as described in Response 3.15. Three unique features of the modeling (highlighted in the contributions paragraph) that may account for the lower solar shares relative to wind and other low-emitting technologies are:

1. High temporal resolution (REGEN captures hourly dynamics for investments and operations)
2. Accelerated decarbonization timeframe to 2035
3. Endogenous end-use decisions and hourly load shapes (which leads to greater electrification, shifts in peak load toward winter, and lower capacity contributions for solar)

We summarize these potential differences in a sentence at the end of the “Drivers of Natural Gas Use in Electric Sector Decarbonization Strategies” section: “Note that the lower solar generation shares are lower than some earlier U.S. decarbonization studies (e.g., [19, 20]) due to our modeling having higher temporal resolution, accelerated decarbonization, and endogenous end-use decisions with hourly load shapes that reflect electrification trends, which have been shown to be associated with lower solar deployment vis-à-vis wind and other low-emitting generation options [14, 21, 22].”

Comment 2.3: “A major concern about this study is that the boundary conditions for the electricity sector are poorly specified. The authors refer in passing to electrification loads related to economy-wide decarbonization in a qualitative way (e.g. line 105), but the composition of load in 2035 and 2050 is not specified in the text or in figures. Perhaps even more important, economy-wide emissions targets are not specified, and emissions figures are not provided, so the amount of decarbonized electricity required for new loads economy-wide can’t be easily deduced.”

Response 2.3: We added Supplemental Information Note S4 to describe end-use results in greater detail, including electricity demand by end use over time and by scenario, hourly load shapes by end use, and economy-wide emissions. We added sentences to the “Modeling Deep Decarbonization in the Electric Sector” section in the main text to highlight the end-use assumptions: “Annual electricity demand and hourly load shapes come from the REGEN end-use model (as described in Methods and Note S1). To reflect the deep decarbonization context for the Net-Zero and Carbon-Free scenarios, the end-use model assumes CO₂ pricing of \$50/t-CO₂ for non-electric sectors beginning in 2025 that increases at 7% per year, which is intended as a proxy for a suite of CO₂ policies for end-use sectors. We conduct sensitivities across different levels of end-use electrification to examine effects on natural gas deployment.” The future work paragraph at the end of the “Discussion” section also mentions: “Finally, these scenarios examined zero emissions power sector goals in the context of deep economy-wide reductions, but overall greenhouse emissions do not reach net-zero levels by 2050. Future work should examine the role of gas across net-zero economy-wide futures.”

Comment 2.4: “This lack of clarity about demand highlights the anomalous result of Figure 12 (also Figures 1A, 18, and 20) which shows 2035 generation across all scenarios being very similar to today’s, and to the presumably low-electrification 2035 reference case. This is not what would be expected in an economy undergoing rapid electrification of end uses in transportation, buildings, production of hydrogen for fuel, etc. Figure 18 shows generation in 2050 that is roughly half the level found in recent net-zero studies. Even though this study is framed as electricity-sector only, it needs at a minimum to explore different levels of demand, since the rest of the economy is an unavoidable boundary condition for the electricity sector and demand would vary widely with economy-wide policies. Almost all of the relevant metrics for the sector – emissions intensity, capacity mix, generation mix, balancing requirements, costs, rate impacts, and resource use – depend on the level of demand and its composition.”

Response 2.4: To examine effects of higher end-use electrification on natural gas deployment, we conducted sensitivities to higher assumed levels of electrification. The results of these experiments are shown and discussed in the subsection “Impact of Alternate Levels of End-Use Electrification.” Additional detail is provided in SI Note S4, which includes a figure showing how our scenarios compare with other economy-wide deep decarbonization scenarios in the literature. Our reference and high electrification scenarios for 2035 and 2050 have similar electricity shares of final energy as other studies in the literature. Our electricity demand growth is on the lower side of the U.S. literature and similar to E.U. net-zero studies, largely due to energy efficiency offsetting some of the growth in electrification. As discussed in detail in Note S4, US-REGEN has endogenous energy efficiency and exogenous end-use technological performance assumptions that lead to lower final energy demand than some other models. Also, some of the very high electricity demand scenarios in the literature are ones that do not have detailed representations of efficiency options and exogenously assume higher shares of electricity-derived synthetic fuels with low roundtrip efficiencies, which lead to higher electricity demand than direct electrification (e.g., for transportation).

Comment 2.5: “Perplexingly, Figure 17 shows the U.S. being a winter-peaking system in 2035 for some reason. This is a major shift from the historical U.S.-wide summer peak. Winter-peaking could be

consistent with a high building electrification load, but if this is the case, why is there only a minimal increase in annual demand?”

Response 2.5: We added Supplementary Figure 34 to SI Note S4 to show hourly load shapes across different regions to illustrate how space heating electrification can raise the ratio of the winter peak to the summer one. Supplementary Figure 33 illustrates how total electricity demand in buildings is relatively modest due to offsetting energy efficiency and due to the fact that, in some instances, heat pumps displace not only natural gas in buildings but also less efficient electrical resistance heating. These insights are consistent with other studies in the literature (e.g., Bistline, et al., 2021; Waite and Modi, 2020). We added text after Figure 1 to explain: “Supplementary Figure 33 and Figure 34 show impacts of electrification on electricity demand and hourly load shapes across these scenarios, which can lead to shifting peak loads toward winter during periods with low solar output in some regions.”

Comment 2.6: “The scale of demand is a key driver of the negative emissions and carbon capture required in the net zero scenarios, including how much biomass fuel for BECCS is required to offset emissions from uncaptured natural gas generation. Despite the key role of BECCS in reaching net zero, the carbon accounting is not shown, nor are the annual biomass requirements for generation fuel, and it is not discussed in the paper how this would compare to sustainable levels of biomass use. The conclusions of this study could look very different if higher electricity demand caused the resulting portfolios to bump up against the limits of carbon capture and negative emissions, due to supply curve steepness or biomass constraints or geologic sequestration limits.”

Response 2.6: Per Response 2.4, the new section on electrification helps to answer these questions. We also added a figure in the SI and text in the body to provide detail on emissions accounting across scenarios and to explain that, “The portfolio of CDR technologies deployed depends on technological cost and availability assumptions as well as the scale of demand (Supplementary Figure 14). BECCS is deployed over DAC through 200–300 Mt-CO₂/yr; however, increasing biomass costs make DAC favorable at the margin for higher demand scenarios (e.g., reaching net-zero with high electrification).”

To provide detail on biomass supply assumptions, we added a sentence to the “Modeling Deep Decarbonization in the Electric Sector” section that “Biomass fuel costs are represented as regional supply curves, which are based on the Forest and Agriculture Sector Optimization Model with Greenhouse Gases (FASOM-GHG) (Supplementary Figure 16).” A figure and paragraph were added to Supplementary Information Note S2 showing these regional supply curves: “Regional biomass supply curves are shown in Figure 16. These curves are based on the Forest and Agriculture Sector Optimization Model with Greenhouse Gases (FASOM-GHG). More information on the forestry and agricultural biomass supply modeling is provided in Appendix B of the US-REGEN documentation.”

To give context on the scale of biomass consumption, we added Supplementary Figure 29 to Note S3 with accompanying text: “Biomass supply and use in net-zero scenarios with reference and high electrification are shown in Figure 29. Biomass consumption from BECCS deployment is 2.46 and 3.06 quads under the net-zero scenarios with reference and high electrification, respectively. Biomass consumption in the U.S. in 2020 totaled 4.5 quads with about 0.4 quads in the electric sector, so BECCS at this scale represents an increase of 45–59% in economy-wide biomass consumption. For context, the Princeton “Net-Zero America” study indicates that, for economy-wide net-zero emissions scenarios, biomass consumption is between 11.7–21.7 quads [26].”

To give context on the scale of CO₂ storage, we added Figure 15 in the Supplemental Information, and some associated text describing the data sources. CO₂ storage capacity is based on NATCARB.

Per Response 3.3, we also conducted several other technology and policy scenarios to understand the sensitivity of the results to assumptions related to BECCS, DAC, and hydrogen and added discussion of these scenarios to the text.

Comment 2.7: “This concern is reinforced by Figure 23, which shows high levels of natural gas use remaining in the overall U.S. economy in 2050. It seems inconsistent to assume U.S. policies that would require zero emissions in electricity while allowing major emissions from natural gas use to continue in the rest of the economy. After all, one of the main motivations driving electricity decarbonization policies is the use of decarbonized electricity to displace fossil fuel use elsewhere in the economy. One concludes either that there is incipient electricity demand that was not incorporated in the authors’ analysis due to limiting the scope of electrification of end uses and the production of decarbonized fuels, or that there is an implicit assumption of very high economy-wide carbon capture and storage or direct air capture (but if so this is also not evident in electricity demand). Note that residual petroleum use and its associated emissions is not addressed, but would further stretch the limits of negative emissions and biomass use.”

Response 2.7: Per Response 2.3, these scenarios are not looking at economy-wide net-zero scenarios. However, the range of economy-wide natural gas consumption reported in 2050 for our deep decarbonization scenarios (15 quads to 24 quads, as shown in Figure 37, Note S3) is consistent with economy-wide net-zero scenarios in the Princeton “Net-Zero America” study, which spans from 0 quads (in the “100% Renewable” scenario) to about 20 quads (in the “Renewable Constrained” scenario). Note that petroleum use is significantly declining in these scenarios due to extensive transport electrification, which drives significant emissions reductions. The text and figures added to SI Note S4 illustrate the extent of these effects. Increasing service demand (e.g., industrial output, heated floor space) are happening simultaneously, which is one reason why natural gas demand does not decline by even more, as described in Response 2.4. Also, the focus of many scenarios in this study is 2035, where economy-wide emissions would not be close to zero even on an economy-wide net-zero path by 2050.

Comment 2.8: “A second concern is a lack of clarity about the modeling and how the modeling might effect the results. A simple example is that the text (line 441 et. seq.) says that the hourly dispatch was done in the static mode, and that the static mode was used only for the years 2035 and 2050. If that is the case, how were the gas capacity factors for every 5 years from 2025 to 2050 obtained?”

Response 2.8: We modified the text description to clarify that the model is run in an intertemporal mode first (with five-year timesteps) for retirement decisions and then for individual five-year periods for investment and operational decisions: “The US-REGEN electric-sector model is run in two modes. In the dynamic mode, the model solves the inter-temporal capacity planning optimization problem across the years 2020–2050 in five-year periods but, due to computational constraints, uses 120 representative hours for each year. This mode cannot capture the hourly operations of short-duration energy storage technologies (e.g., lithium-ion battery storage), though longer-duration options are included. In the static mode, a shorter foresight optimization is cast as a single year static equilibrium model with capacity investment and hourly dispatch using the retirement information from the intertemporal solve. The use of static mode allows the analysis to represent hourly operations and capacity investments faithfully, something that is not currently possible in large-scale intertemporal models [36]. The literature has demonstrated that a model must be able to capture the declining economic value of variable renewable energy at higher penetration levels and ability of system resources like energy storage to mitigate these effects, which are captured in US-REGEN [15, 27]. This formulation is conceptually similar to a sequential myopic model [37].”

We also believe that making the model code available upon publication will provide additional clarity.

Comment 2.9: “A more important question not answered in the methodology descriptions is whether the US-REGEN model has the ability to represent sector coupling between the electricity and fuels production sectors. If not, it is difficult for the authors model to accurately represent effects that have a strong influence on the central tradeoffs being explored in this paper, for example battery storage vs gas capacity, the cost of producing hydrogen by electrolysis, etc. In this regard, the modeling may not have the capacity to give confidence regarding some of the claims of the paper, and the authors should provide stronger caveats.”

Response 2.9: We added paragraphs to the “Methods” section to describe the sector coupling in US-REGEN: “The US-REGEN end-use model captures intensive and extensive margin responses through investment and operational decisions. The model focuses on representing drivers for electrification from customer and firm perspectives, with considerable heterogeneity across households, industries, and regions, and is unique in its simulating economic and behavioral factors for end-use decisions rather than specifying adoption decisions exogenously as model inputs [32]. Non-electric sectors are assumed to face a carbon price as a proxy for decarbonization incentives at the end-use level, which starts at \$50/t-CO₂ in 2025 and increases at 7% per year. The end-use module includes a module to value and project investments in residential and commercial rooftop solar, and another module that allows for the opportunity to defer electric vehicle charging to reduce or defer peak demand.¹ The end-use model generates hourly electricity load profiles by region which are inputs to the electric sector model.

The detailed electric sector capacity expansion and end-use models are run iteratively, with the electric model passing hourly electricity prices to the end-use model, and the end-use model passing back hourly load shapes and load growth, until energy prices and demands converge between the two models (Supplementary Figure 7). In addition to electricity load shape flexibility, there is sector coupling between electricity and fuels production, including endogenous hydrogen production with range of pathways, as described in Note S2.”

US-REGEN endogenously computes demand for electricity, natural gas, refined petroleum, and hydrogen. The model exogenously fixes the price of coal, gas, and petroleum, but endogenously computes the supply (and the cost thereof) for hydrogen and electricity. So hydrogen and electricity are fully coupled. The markets for coal and refined petroleum are not pertinent to the research question addressed here, but the supply and cost of natural gas is obviously a critical sensitivity, which is the reason we run high natural gas price sensitivities to ensure that we capture the potential impact to the results.

Comment 2.10: “Even without knowing the intimate details of the model, there is evidence to support this concern. The low values for solar throughout indicate that load flexibility is limited in the modeling –the ability to do flexible charging or to shift to large industrial flexible loads like electrolysis seems to be low; if not, there would be more solar and more hydrogen production. (It appears that “green hydrogen” in the fuel mix is exogenous, not an endogenous result of sector coupling). This could be due to the apparent low level of electrification mentioned earlier, or to inability to model sector coupling.”

Response 2.10: Per Response 2.9, we added text to the Methods and SI to describe the assumptions about sector coupling, load flexibility, and fuels production (including hydrogen). Note that hydrogen production (blue and green) is endogenous, and is predominantly used in the power sector here and is less competitive for end-use decarbonization due to lower costs of electrified alternatives, unless scenario assumptions specifically constrain electrification and CCS, as studies with higher hydrogen production tend to be ones that exogenously assume end-use will use hydrogen instead of electricity (Williams, et al., 2021; Bistline, 2021). To be sure, we would expect to see less hydrogen regardless given the focus on our

¹ The participation share in charging flexibility programs is assumed to be 50% for residential households and 80% for workplace charging for this analysis.

paper on 2035 electric sector decarbonization and the fact that hydrogen tends to emerge only as economy-wide emissions are nearing net-zero levels.

Sector coupling, especially with deferrable vehicle charging, can be a substitute for utility-scale energy storage, which means that the impact of sector coupling on specific assets is context-dependent and prima facie ambiguous. Lower solar in our scenarios is likely due to higher temporal resolution and focus on 2035 decarbonization. We added text after Figure 1 to describe potential differences: “Note that solar generation shares are lower than some earlier U.S. decarbonization studies (e.g., [19, 20]) due to our modeling having higher temporal resolution, accelerated decarbonization, and endogenous end-use decisions with hourly load shapes and electrification, which can be associated with lower solar deployment vis-à-vis wind and other low-emitting generation options [14, 21, 22]. Supplementary Figure 33 and Figure 34 show impacts of electrification on electricity demand and hourly load shapes across these scenarios, which can lead to shifting peak loads toward winter during periods with low solar output in some regions.”

Comment 2.11: “One speculates that the latter might be because the US-REGEN model does not have the ability to represent cumulative energy surpluses and deficits (aka state of charge) across all hours of the year (see e.g. “Optimal gas-electric energy system decarbonization planning” by Von Wald et al 2022, starting on p.4, for a recent treatment of this question), which is necessary to represent sector-coupling accurately.”

Response 2.11: US-REGEN has hourly resolution on the supply and demand side of the model, so can and does track cumulative energy surpluses and deficits, which is one unique feature making it very well-suited for an analysis like this one. We added text to the “Introduction” to highlight this feature: “Modeling electric sector investment and operational decisions with full hourly temporal resolution, endogenous end-use decisions and load shapes, as well as a greater suite of technological options to better represent the economic characteristics of variable renewables, energy storage (both short-duration options like batteries and longer-duration ones like electrolytic hydrogen), and dispatchable low-carbon technologies. Hourly resolution is important not only for accurately characterizing the investment and operations of electric sector resources but also for capturing sector coupling dynamics such as load flexibility and fuels production.”

Comment 2.12: “Several of the results figures, for example Figures 4, 5 and 15, show such wide ranges that they are not really informative and one can draw few definitive conclusions from them. This problem is not confined to the present manuscript, but is found too widely in energy modeling studies. It is nonetheless not of the standard that would be expected in a Nature journal.”

Response 2.12: We removed Figure 5 from the Discussion section, instead opting to make qualitative comparisons in the text. Additionally, we updated the prior Figure 15 (which is now Figure 22 in the updated text) to separate columns to differentiate between reference and zero-emissions scenarios. We kept Figure 4 as it is, segmenting results by scenario, which helps to diagnose drivers. The range is indicative of underlying uncertainty and differences across regions, and this nuance is important not to artificially diminish.

Comment 2.13: “The generic use of “gas” as a substitute for “natural gas” for brevity (line 26) is problematic. The authors themselves do not adhere strictly to this convention, and this usage is confusing in situations in which there could be expected to be gas blends with varying levels of fossil natural gas.”

Response 2.13: We revised the manuscript to use natural gas, hydrogen (differentiating between green and blue hydrogen, where appropriate), and other more specific terms throughout.

Comment 2.14: “Some of the writing in the paper is less clear than it could be. An example is the last sentence of the abstract.”

Response 2.14: We rephrased the last sentence of the abstract to clarify: “Nonetheless, wind and solar have higher generation shares than natural gas for many regions and scenarios (45% to 66% variable renewables for net-zero scenarios versus 3% to 26% for gas).”

Comment 2.15: “There is an “error reference not found” at line 261.”

Response 2.15: We fixed this reference error.

Comment 2.16: “It is difficult to recommend publication of this manuscript in a Nature journal. The fact that the top line results support findings that have been found in other recent studies regarding the value of gas generating capacity in maintaining reliability in a net zero electricity system does not redeem the paper’s other flaws. It is not a novel result to conclude that inexpensive natural gas coupled with seemingly unlimited negative emissions capacity will favor a net-zero scenario over a carbon-free scenario in an optimization model. If the paper were returned to the authors for a rewrite, their main task would be to address the lack of clarity about boundary conditions regarding emissions and electricity demand through new model runs, new scenarios, and a thorough re-write, with much greater detail about what is assumed for the rest of the economy, including economy wide energy mixes and emissions, including biomass use, negative emissions, and CCS.”

Response 2.16: We trust that our revisions, expanded scenarios, and new figures all help to provide clarity for readers and provide thorough responses to the reviewer’s feedback. As the reviewer notes in Comment 2.1, our paper informs a number of important policy debates about electric sector and economy-wide decarbonization, which are extremely timely in light of U.S. utility decisions about near-term investments. We believe that our paper provides several methodological and analysis improvements over earlier work:

- Modeling electric sector investment and operational decisions with full hourly temporal resolution, endogenous end-use decisions and load shapes, as well as a greater suite of technological options. Hourly resolution is important not only for accurately characterizing the investment and operations of electric sector resources but also for capturing sector coupling dynamics such as load flexibility and fuels production.
- Including a wide range of sensitivities, including being the first peer-reviewed article to assess impacts of accelerating zero-emissions goals to 2035, per the updated U.S. Nationally Determined Contribution.
- Evaluating the role of natural gas in a range of regional power system contexts with different existing capacity mixes, natural gas prices, renewable resources, and demand characteristics.

Reviewer #3

Comment 3.1: “The authors analyze the role of natural gas in combination with carbon capture, sequestration and removal technologies in decarbonization scenarios for the United States. To do so, they use an optimization model that has been developed by EPRI and that has already been applied in previous research. Their main finding is that natural gas plays a substantial role in all investigated decarbonization scenarios under the assumptions made here. The only exception is a “pessimistic” sensitivity where several assumptions are combined that make power generation from natural gas unfavorable. The authors also highlight that wind and solar have higher generation shares than natural gas in all scenarios analyzed.”

Response 3.1: Thank you for your encouraging comments and constructive feedback, which have been helpful as we refined the manuscript.

Comment 3.2: “In general, I think the main part of the analysis and the major conclusions drawn are not so much about natural gas (or its “system value”) per se, but rather about gas in combination with CCS technologies, including carbon removal via BECCS or DAC. So an alternative (and maybe more appropriate) title of the paper may be something like “how carbon capture and removal can help in achieving net zero emissions in the electric sector”. In the Discussion, the authors state “Our results indicate that new and existing gas can help to reduce emissions...” – but in fact, it is predominantly CCS/BECCS that facilitates net-zero emissions here, and not so much natural gas as such.”

Response 3.2: We altered the title to include carbon removal and to emphasize the geographical focus of the analysis: “Is There a Role for Natural Gas and Carbon Removal in Reaching Net-Zero Emissions in the United States Electric Sector?” We expanded the Discussion sentence to: “Our results indicate that new and existing gas—enabled by carbon removal via BECCS or DAC as part of a net-zero electric sector—can help to reduce emissions, facilitate dependable system operations, and reduce the cost of decarbonizing the electric sector.”

Comment 3.3: “Yet, I find it hard to fully grasp the role of natural gas in combination with CCS/DAC technologies, as there is no alternative scenario in which BECCS/DAC are fully excluded (also not in “NoGasCCS”). In such a setting, alternative decarbonization strategies such as combinations of renewables and storage would have to be used. There is only the “pessimistic” scenario, but here, several sensitivities are combined together, and CCS is still not fully excluded. At least from a European perspective (which I have), the assumption that no BECCS/DAC may become available for the power sector appears to be policy relevant. In fact, several European countries currently struggle with the development of CCS infrastructure to deal with hard-to-abate industrial / process emissions – let alone develop additional CO₂ transportation and storage infrastructure for CO₂ from gas-fired power plants. This may be different in the US – but still, the effect of gas+CCS could be better quantified against a reference without any CCS. So in my view it would strengthen the validity and robustness of results if there was a setting without any CCS/BECCS/DAC technologies. This would also support the statement “Natural gas’ role in a net-zero system hinges on carbon capture...” (line 208).”

Response 3.3: We included an alternative scenario along the lines the reviewer suggests with BECCS/DAC fully excluded, which we call the “Carbon-Free” scenario. This scenario was not included in the first couple figures in the earlier draft, so in this revision, we added it as a bookend to the “Net-Zero” scenarios. We also added a paragraph and figure in Note S2 to describe the CO₂ transport and storage formulation and assumptions in the model.

Additionally, we conducted several other technology and policy scenarios to understand the sensitivity of the results to assumptions related to BECCS, DAC, and hydrogen. These scenarios were added to SI Note

S3 and with a figure (Supplementary Figure 28). This description starts with the text: “To test the robustness of results to alternate technological assumptions...” We reference these sensitivities in the main text just after Figure 1: “BECCS is the main CDR technology used to reach Net-Zero goals due to its lower cost of net CO₂ removal and provision of firm negative-CO₂ electricity; however, DAC deployment can increase under alternate technological cost and availability assumptions (Supplementary Figure 28).”

Per Response 3.12, we also added a paragraph to the “Discussion” section to highlight potential differences between the U.S. setting examined in the paper and other geographies, specifically calling out challenges in Europe associated with CO₂ transport and storage infrastructure: “These insights focus on the potential role of natural gas in the U.S. electric sector. Several unique features about the U.S. setting may make insights less transferrable to other country contexts, including its lower-cost fossil fuel resources, plentiful biomass, high-quality wind and solar resources, and ample CO₂ sequestration nationally. Each of these features exhibits regional heterogeneity across the country that can give rise to variation in the competitiveness of natural gas (Figure 4). Countries like the European Union not only have higher costs for natural gas and other fuels but also may see more limited CCS deployment due to infrastructure challenges associated with CO₂ transportation and storage.”

Comment 3.4: “A related point: I guess that the cost assumptions for CCS/BECCS/DAC technologies are relatively favorable compared to renewable costs, judging from Figure 8 and from the relatively low optimal renewable shares in decarbonization scenarios. Yet, DAC and carbon storage costs are not shown in Figure 8, and they are also not provided elsewhere as far as I can see (maybe I missed it).”

Response 3.4: Per Response 1.6, we added text, figure, and table to Supplemental Information Note S2 to describe the representation of DAC, parametrization, and data sources. We added a paragraph and figure in Note S2 to describe the CO₂ transport and storage formulation and assumptions in the model.

Comment 3.5: “Further, I think the parameterization of scenarios and sensitivities could be revisited. For example, I don’t see why the reference assumes zero upstream methane leakage – this is clearly unrealistic. I guess the “1.5% Leak” sensitivity could be the baseline assumption. Likewise, it is not clear if some long-term emissions from captured/sequestered carbon are accounted for in the analysis. I am not an expert in this field, but the issue of long-term CO₂ leakage from storage/sequestration could at least be discussed (or represented in the parameterization, if relevant).”

Response 3.5: Per Response 1.8, we reran all of the scenarios with a 1.5% upstream CH₄ leakage rate as the default assumption with 3% as the sensitivity. We added a sentence to SI Note S3 that, “We assume that all captured CO₂ is permanently sequestered [41].”

Comment 3.6: “Next, there are several statements in the manuscript which appear somewhat bold and/or under-developed to me. For example, the authors claim that natural gas is relevant “not only during the transition to net-zero emissions but also at the destination” (lines 100-101). Yet, hardly any results are provided for the transition part of this statement, the focus is clearly on the 2035 “destination”. ”

Response 3.6: We added a sentence to this paragraph to clarify: “Supplementary Figure 19 shows how coal generation shares rapidly decline across all regions in emissions-constrained scenarios, while natural gas shares exhibit much slower declines.” We added Supplementary Figure 19 in SI Note S3 to illustrate the transition dynamics for coal and natural gas generation.

Comment 3.7: “Likewise, the statement “The extent of gas generation for a decarbonized electric sector depends on policy design, ability to mitigate upstream methane, and transition risks from technological change” (107-108) is extremely broad and not fully connected to the results presented.”

Response 3.7: We extended this paragraph and subsequent ones to provide more direct support for the claims in the topic sentence. Overall, our approach is to start with broader summary sentences and then provide more detailed supporting information beneath such topic sentences.

Comment 3.8: “Another example: “...least-cost decarbonization pathways include emerging technologies such as advanced nuclear, hydrogen, and capacity equipped with carbon capture and sequestration” (110-112): this is a very general statement that appears to be not really supported by the results presented. CCS in fact is always a relevant part of the solution in the results presented (except for “Pess”), but advanced nuclear and hydrogen are relevant only in very specific cases.”

Response 3.8: We agree that this sentence was too general and removed it.

Comment 3.9: “To give another example, I find it hard to relate the statement made on line 136 “Accelerating decarbonization entails greater contributions from gas on a relative and absolute basis...” to the material presented in Figure 4.”

Response 3.9: We added Figure 5, which compares 2035 and 2050 decarbonization target scenarios. In the subsequent sentence, we also reference Supplementary Figure 24 that compares generation mixes in the 2035 and 2050 policy scenarios. These figures do a better job of providing support for the claim that “Accelerating decarbonization entails greater contributions from gas on a relative and absolute basis...”

Comment 3.10: “Finally, the authors further seem to suggest that decarbonization gets cheaper if we wait longer, as renewable energy costs will by assumption decrease over time (cp. Lines 159 ff.). I think this may lead to potentially flawed conclusions. This conclusion is clearly driven by the assumption that renewable get cheaper over time, irrespective of the speed of deployment (in the US and globally). Further, the IPCC has repeatedly stated that the costs of climate change increase, the longer we wait with mitigation. Maybe the authors don’t mean to suggest this, but the reader may interpret these statements such that delaying decarbonization ambitions would be preferable.”

Response 3.10: We added a sentence to the text to highlight this important caveat: “However, note that electricity prices only track system costs and do not explicitly include monetized estimates of climate damages avoided from lower CO₂ emissions or co-benefits such as human health benefits from air quality improvements [23], which would be higher under nearer-term decarbonization pathways.” We also added a caveat in the caption of Figure 3: “Estimates reflect electric sector system costs as a proxy for customer impacts and do not include climate damages or other social costs or benefits.”

Comment 3.11: “In general, I think the research question is timely and policy relevant.”

Response 3.11: We thank the reviewer their assessment of the importance of the issues addressed.

Comment 3.12: “Yet, at least from an EU perspective, the reader probably would expect some other insights from a paper with this title. In Europe, we currently have a fierce debate about the role of natural gas for the decarbonization of the energy system. This recently became very visible in the context of the taxonomy debate, i.e. can investments in natural gas power plants considered to be sustainable or not. An important question here is not only for how long natural gas can be used for generating electricity, but also how and when it could be substituted partly or completely by (low-carbon) hydrogen. By design, the analysis is completely silent on this aspect, as it allows the use of natural gas also in the future via CCS/BECCS/DAC.”

Response 3.12: To underscore which insights from the paper are less transferable outside of the U.S. context, we added a paragraph to the “Discussion” section to highlight unique features of the U.S. relative to other countries: “These insights focus on the potential role of natural gas in the U.S. electric sector. Several unique features about the U.S. setting may make insights less transferrable to other country contexts, including its lower-cost fossil fuel resources, plentiful biomass, high-quality wind and solar resources, and ample CO₂ sequestration nationally. Each of these features exhibits regional heterogeneity across the country that can give rise to variation in the competitiveness of natural gas (Figure 4). Countries like the European Union not only have higher costs for natural gas and other fuels but also may see more limited CCS deployment due to infrastructure challenges associated with CO₂ transportation and storage.” The title and abstract were modified to emphasize the U.S. context here.

We also added several sensitivities related to the interplay between natural gas, carbon removal, and hydrogen to SI Note S3 with the text beginning: “To test the robustness of results to alternate technological assumptions...” We believe that these scenarios offer insights into the competitiveness of hydrogen vis-a-vis natural gas in the power sector in the U.S.

Comment 3.13: “The finding that renewables have higher generation shares than natural gas in cost-optimized decarbonization scenarios is not very surprising and also not exactly new. In fact, this may be considered a well-established fact that has been demonstrated in various decarbonization studies in the literature.”

Response 3.13: We modified the wording of this finding in the Discussion section to make reference to the alignment in the conclusion with the literature, though noting differences in methods² between our study and earlier ones: “The extent of deployment and utilization of gas depends on policy, technology, and market uncertainties. Wind and solar exhibit greater increases in generation shares for many regions and scenarios, especially with stringent CO₂ policies (52% to 66% variable renewables for net-zero scenarios versus 0% to 19% for natural gas). These findings generally agree with earlier studies of U.S. decarbonization [23, 20, 19, 24, 25]—albeit this study looks at deeper decarbonization goals, lower renewable costs, and greater variety of sensitivities using a model with hourly temporal resolution and endogenous load shapes. These differences generally mean that deployed natural-gas-fired capacity is on the lower side of existing multi-model deep decarbonization scenarios [9] and economy-wide net-zero studies [20, 26] due to the greater number of scenarios investigated here (including some with pessimistic assumptions about gas and optimistic ones about other technologies).”

We feel that it is important to underscore this point about high renewable shares in the abstract, introduction, results, and discussion to ensure that readers don’t lose sight of this large role of wind and solar despite other results that focus on the potential roles of gaseous fuels and carbon removal.

Comment 3.14: “The authors provide a high-level description of the model they use, and refer to a more detailed model description published by EPRI. The model itself is not provided open-source as far as I can see (or did I miss it?). Especially when it comes to the concrete input parameters used, this leads to a lack of transparency, and it impedes reproducibility. I think it is fair to consider the open-source provision of models and input data as best practice in the energy modeling field, and I personally would also expect this from research published in a highly ranked journal such as Nature Communications.”

Response 3.14: []. We added a “Code Availability” statement to the end of the article: “The optimization code that supports the analysis within this paper is available from the corresponding author upon reasonable request.”

² Per Comments 2.2 and 2.5, our modeling differs from other studies in the mix of renewables and energy storage deployment in part due to the higher temporal resolution and endogenous load shapes.

Comment 3.15: “The renewable cost assumptions appear to be rather pessimistic compared to previous analysis, especially for solar PV. For example, compare the analysis by Victoria et al. published in the same journal (<https://doi.org/10.1038/s41467-020-20015-4> [doi.org]), where renewable costs are much lower. Looking at the numbers provided there, the “LoRE” sensitivity could probably be considered as the baseline here. In turn, the CCS / BECCS / DAC cost assumptions are probably relatively optimistic (but they are not fully transparent, see remark above).”

Response 3.15: We adjusted our solar PV, onshore wind, and offshore wind costs, which are closer to our previous “LoRE” sensitivity. Figure 11 in Note S2 shows these costs and compares them with other U.S. projections³ in the literature.

We also add lower and higher renewable cost scenarios where we vary solar PV and wind individually to illustrate the sensitivity of the results to alternate cost assumptions. The text and a new figure were added to the body of the paper in a section called “Sensitivity to Wind and Solar Costs.”

Per Response 2.6, higher CCS costs and CDR costs are addressed in additional scenarios. As the choice set of firm technologies is constrained or made more costly, the generation and capacity mix moves closer to the “Carbon-Free” scenario where carbon removal and natural gas are prohibited.

Comment 3.16: “Are there variable costs of bioenergy? In fully decarbonized energy systems, it is quite plausible that bioenergy is scarce (and thus, expensive), as it is likely to be used in the transportation sector (and maybe for heating). Another question is if the additional load required for DAC is accounted for in the model. At least this is not visible in Figure 1.”

Response 3.16: Per response 2.6, we added a sentence to the “Modeling Deep Decarbonization in the Electric Sector” section that “Biomass fuel costs are represented as regional supply curves, which are based on the Forest and Agriculture Sector Optimization Model with Greenhouse Gases (FASOM-GHG) (Supplementary Figure 16).” A figure and paragraph were added to Supplementary Information Note S2 showing these regional supply curves and a figure to show regional biomass consumption for these scenarios relative to these supply curve steps. DAC demand is shown in SI figure on electricity demand and assumptions about DAC cost and performance are now provided in Note S2.

Comment 3.17: “Concerning the methodology, I did not fully understand to what extent the model is dynamic. Of course, a dynamic model would be very desirable for this kind of analysis, and would distinguish this analysis from several previous papers. Yet, the Supplemental Information suggests that the dynamic mode is used only for retirement of existing capacity. But again, this is hard to check as model code is not provided.”

Response 3.17: Per Response 2.8, we rewrote this paragraph to clarify the degree of foresight used for retirement decisions versus investment and operational ones. We agree that the code will provide an additional layer of clarity for audiences who are interested in the implementation.

Comment 3.18: “Likewise, it became not clear to me if capacity decisions related to both energy and power are endogenous only for batteries, as suggested in the respective paragraph in the SI, or also for other storage technologies.”

³ Note that the solar PV values used in the paper may differ from non-U.S. studies owing to country-specific differences in costs. Also, we show costs in $\$/kW_{AC}$ terms, which makes values seem higher than solar costs presented in $\$/kW_{DC}$ terms.

Response 3.18: We added a sentence in the “Methods” section to clarify: “Energy capacity and power capacity are endogenously optimized for all energy storage technologies in the model.”

Comment 3.19: “Regarding the electricity prices shown in Figure 3: Here the reference has the lowest prices, and decarbonization scenarios are always more expensive. When prices are interpreted as an indicator of “costs”, which seems to be the case here, this may lead to flawed conclusions, as there are substantial external costs related to non-abated carbon emissions in the reference. If I understand correctly, CO2 is not priced in the reference.”

Response 3.19: Per Response 3.10, we added a sentence to this section and one to the caption of Figure 3 to convey these important caveats. We agree that emphasis should be on comparing costs of pathways with identical emissions trajectories (e.g., Net-Zero vs. Carbon-Free for particular years), as we attempt to do in the text.

Comment 3.20: “I think the scenario space of this study could be reduced. The authors introduce three policy targets (reference, carbon-free and net-zero), as well as two target years (2035 and 2050). Yet, most of the presented results focus on the net-zero 2035 setting. So it would be clearer and less confusing to make this focus more clear already in the beginning of the results section.”

Response 3.20: The revised version of the manuscript places more emphasis on the 2035 target year, but we feel that keeping the 2050 target year in the analysis is important, given how many U.S. utilities currently have 2050 net-zero targets, even though the Biden Administration has set a 2035 net-zero goal. We feel that the large number of scenarios is a virtue of the study to answer many of the specific questions decision-makers have as well as to illustrate which points are robust across a wide range of possible policy, technology, and market uncertainties.

Comment 3.21: “I generally like the idea of comparing the results to other studies, but the respective paragraph in the Discussion section (including Figure 5) is underdeveloped in my opinion. For readers not very familiar with the other studies mentioned, it is very hard to understand Figure 5 and the different results provided for 2030 (?) and 2050.”

Response 3.21: We agree with the reviewer’s points and have removed Figure 5 from the Discussion section, instead opting to make qualitative comparisons with other studies in the text.

Comment 3.22: “Line 27: the statement seems to be a fairly general one, but both references only refer to the U.S.”

Response 3.22: We added “...in the U.S.” to this statement.

Comment 3.23: “Line 38 ff: The contributions may be better placed after the following paragraph where the research question is stated.”

Response 3.23: We moved the sentence “Our objective is to assess the potential role for natural gas in deeply decarbonized electricity systems in the U.S. and evaluate the robustness of this role to key technology and policy assumptions.” before the bulleted list of contributions.

Comment 3.24: “Is it plausible that an increase in the non-electric gas use would be allowed in the carbon-free scenario (lines 157ff.)?”

Response 3.24: We added a footnote to this sentence: “Note that this emissions rebound effect could be mitigated if a quantity-based economy-wide emissions policy (e.g., cap-and-trade) were used as the primary policy instrument instead of a price-based CO₂ policy.”

Comment 3.25: “The “Regional Differences in Decarbonization Strategies” section appears to be quite US-specific. It is not clear to me if the general findings could also be transferred to other world regions.”

Response 3.25: Per Response 3.12, we added a paragraph to the “Discussion” section to underscore unique features of the U.S. context that make some of the paper’s findings less transferable to other regions. We also modified the introductory sentence in the “Regional Differences in Decarbonization Strategies” section to point out that these are U.S. regions: “There are important differences in the competitiveness of natural gas across U.S. regions.”

Comment 3.26: “Figure 4 looks nice, but I find it hard to understand. Especially, the differences between 2035 and 2050 are hard to see. I am also not sure how to interpret results because of the aggregation (“Shares include natural-gas fired generators with and without carbon capture as well as hydrogen through steam methane reforming”).”

Response 3.26: We added text near this figure to make the insights easier to understand.

Comment 3.27: “In the Discussion section, it is argued that natural gas would be compatible with net-zero goals if “(2) the electricity market structure effectively values capacity in high-renewables systems”. I am not sure that this can be considered a direct outcome of the study; it rather seems to be an implicit conclusion (as capacity is actually valued in the optimization model).”

Response 3.27: We removed this text from this sentence.

Comment 3.28: “Onshore / offshore wind are different technologies with different costs, but results are only shown for “wind” (maybe no offshore in the optimum?).”

Response 3.28: We added a footnote toward the end of the “Drivers of Natural Gas Use in Electric Sector Decarbonization Strategies” subsection to explain: “Note that figures aggregate onshore wind and offshore wind into a single category, since offshore wind capacity is driven primarily by state mandates (approximately 32 GW by 2035) and does not vary considerably across scenarios.” We also include a sentence in the Methods section on offshore wind mandates: “Offshore wind mandates in CT, MA, MD, ME, NJ, NY, RI, and VA based on legislation in those respective states.”

Comment 3.29: “Very recently, the Russian attack on the Ukraine has intensified the question how to substitute natural gas in the European power sector. The analysis could probably connect to this development and discuss the implications.”

Response 3.29: Per Response 3.12, the analysis focus on the U.S. makes us hesitant to draw firm conclusions about the transferability of these insights into other settings. Although the work sheds light on how the competitiveness of natural gas varies when prices are higher and alternative assumptions about hydrogen and carbon removal are made, we hesitate to speculate on how our results for the U.S. might link to the E.U. setting.

Comment 3.30: “The references appear to be somewhat unbalanced in my opinion. The manuscript includes 38 references, of which one is mentioned twice, so 37 unique references overall. Of these, 13 are grey literature, and 24 are peer-reviewed. The peer-reviewed articles are heavily focusing on the US. I think it would be reasonable to also connect to related model analysis from other world regions. In

particular, many relevant model-based decarbonization studies have been published for Europe, which should not be ignored here. What is more, of the 24 peer-reviewed references, 12 are self-citations. I fully appreciate that the authors have made important contributions to the field before – but I think this is an excessive level of self-citations.”

Response 3.30: We appreciate this gentle note on the optics of our references! We revised the citations to include more papers not written by us and to delete several citations to our own work.

Reviewer comments, second round

Reviewer #1 (Remarks to the Author):

The manuscript has been greatly improved. I appreciate the authors thoroughness in addressing my comments on the initial submission. I recommend this article for publication.

Reviewer #2 (Remarks to the Author):

I have read the revised manuscript and the authors' responses to the reviewers, and in my view the revised manuscript merits publication in Nat Comm. I appreciate that the authors have clearly worked hard on the revision and taken the reviews seriously. They successfully responded to the concerns raised in my review through additional modeling using different assumptions, improved figures, and new explanatory text regarding electricity demand scenarios, economy wide emissions assumptions, renewable vs natural gas generation share, carbon capture, and biomass use. This article will be a valuable contribution to the literature.

Reviewer #3 (Remarks to the Author):

I would like to thank the authors for kindly replying to all of my comments and for addressing most of them in the revised version of the manuscript. I appreciate the additional work the authors have put into the revision and think the manuscript has benefited substantially, also guided by the comments made by reviewers #1 and #2.

There is one point that still concerns me: In the year 2022, with all the discussions about open science and reproducibility we had in recent years, I think it would really be required to provide the code and the data used in this study open-source, if this manuscript is to be published in a top journal as this one. This means not only providing the code and data "upon reasonable request", but to provide it freely and without any barriers in a public repo, using a permissive license. I know this is, unfortunately, still not a standard procedure in energy modeling - but it should be, and we should expect it from the very top journals as NCOMMS in my opinion. I'm sure the authors will find a way to share code and data without making their employer too unhappy (assuming that this is the major barrier).

I have only three minor remaining points:

- the European Union is not a "country", but, well, a union of individual countries. So please rephrase the new sentence "Countries like the European Union..."
- I'm still not entirely convinced about Figure 4. I guess there must be some other way of showing the different regional distributions of results than using such an unstructured scatter plot? Maybe also including some regional information?
- I still think it would be interesting to add at least some discussion on the question if existing and/or new natural gas plants could make a fuel switch to hydrogen in the future, and at which costs (compare my previous comment 3.12).

Response to Reviewers

Reviewer #3

Comment 3.1: “I would like to thank the authors for kindly replying to all of my comments and for addressing most of them in the revised version of the manuscript. I appreciate the additional work the authors have put into the revision and think the manuscript has benefited substantially, also guided by the comments made by reviewers #1 and #2.

There is one point that still concerns me: In the year 2022, with all the discussions about open science and reproducibility we had in recent years, I think it would really be required to provide the code and the data used in this study open-source, if this manuscript is to be published in a top journal as this one. This means not only providing the code and data "upon reasonable request", but to provide it freely and without any barriers in a public repo, using a permissive license. I know this is, unfortunately, still not a standard procedure in energy modeling - but it should be, and we should expect it from the very top journals as NCOMMS in my opinion. I'm sure the authors will find a way to share code and data without making their employer too unhappy (assuming that this is the major barrier).”

Response 3.1: We plan to release the data and code publicly once the embargo has been lifted for our article. We modified the code availability accordingly: “The optimization code and data that support the analysis within this paper are available from the repository <https://github.com/b3311/gasnetzero>.”

*Comment 3.2: “I have only three minor remaining points:
- the European Union is not a "country", but, well, a union of individual countries. So please rephrase the new sentence "Countries like the European Union...".”*

Response 3.2: We rephrased this sentence to: “Countries like those in the European Union...”.

Comment 3.3: “- I'm still not entirely convinced about Figure 4. I guess there must be some other way of showing the different regional distributions of results than using such an unstructured scatter plot? Maybe also including some regional information?”

Response 3.3: We updated Figure 4 to show regional results as different shapes.

Comment 3.4: “- I still think it would be interesting to add at least some discussion on the question if existing and/or new natural gas plants could make a fuel switch to hydrogen in the future, and at which costs (compare my previous comment 3.12).”

Response 3.4: We added a paragraph just before the “Sensitivity to Wind and Solar Costs” section on hydrogen cofiring, blending, and conversion at existing and new gas-fired plants: “Companies have announced plans to cofire or blend hydrogen at existing and new natural-gas-fired plants or to fully convert these plants in the future, and gas turbine manufacturers are designing equipment to handle large shares of hydrogen [24]. However, hydrogen generation shares are modest across many scenarios in this analysis due to their higher marginal abatement costs. For instance, for \$1/kg hydrogen (roughly the current costs of production with steam methane reforming or 2050 costs with electrolysis according to BloombergNEF [25]) and \$4/MMBtu natural gas, abatement costs of hydrogen cofiring, blending, or conversion are about \$90/t-CO₂ before accounting for upstream emissions associated with hydrogen production [26]. At these marginal abatement costs, power sector CO₂ emissions can be lowered 90-95%

from 2005 levels [18], indicating that natural gas to hydrogen fuel switching would not be economic unless lower relative costs were achieved.”